



**Parameterized reactivity of hydroxy radical, ozone, nitrate radical and**
**atmospheric oxidation capacity during summer at a suburban site between Beijing**
**and Tianjin**
Yuan Yang[1,2], Yonghong Wang[3], Dan Yao[1,2,6], Dongsheng Ji[1], Jie Sun[1], Yinghong Wang[1], Shuman
Zhao[1,2], Wei Huang[1,2], Shuanghong Yang[1,5], Wenkang Gao[1], Zirui Liu[1], Bo Hu[1], Renjian Zhang[1],
Limin Zeng[4], Tuukka Petäjä[3], Veli-Matti Kerminen[3], Markku Kulmala[3], Yuesi Wang[1,2,6]
[1] Institute of Atmospheric Physics, Chinese Academy of Sciences, Beijing 100029, China
[2] University of the Chinese Academy of Sciences, Beijing 100049, China
[3] Institute for Atmospheric and Earth System Research / Physics, Faculty of Science, P.O.Box 64,
00014 University of Helsinki, Helsinki, Finland
[4] State Joint Key Laboratory of Environmental Simulation and Pollution Control, College of
Environmental Sciences and Engineering, Peking University, Beijing 100871, China
[5] Department of Environmental Science and Engineering, Beijing University of Chemical
Technology, Beijing 10029, China
[6] Center for Excellence in Regional Atmospheric Environment, Institute of Urban Environment,
Chinese Academy of Sciences, Xiamen 361021, China
Submitted to:Atmospheric Chemistry and Physics
Corresponding to: Yonghong Wang, yonghong.wang@helsinki.fi;
Yuesi Wang, wys@mail.iap.ac.cn

28 :



**Abstract**
Hydroxyl (OH) radicals, nitrate ($NO_3$) radicals, and ozone ($O_3$) play central roles in the troposphere
because they control the lifetimes of many trace gases that resulted from anthropogenic and biogenic
origins. To estimate the self-cleaning capacity of the atmosphere, the reactivities of OH, $NO_3$ and
$O_3$ were comprehensively analyzed based on a parameterization method at a suburban site in
Xianghe in the North China Plain from 6 July 2018 to 6 August 2018. The site had suffered the most
abundant annual mean VOCs concentrations according to a network observation from 2012-2014
(personal communication). The total OH reactivity, $R_{OH}^{total}$, $NO_3$ reactivity, $R_{NO_3}^{total}$, and $O_3$ reactivity,
$R_{O_3}^{total}$, at the site varied from 8.5 $s^{-1}$ to 68.1 $s^{-1}$, 0.7 $s^{-1}$ to 27.5 $s^{-1}$, and 3.3×10$^{-4}$ $s^{-1}$ to 1.8×10$^{-2}$ $s^{-1}$
with campaign-averaged values of 25.6±9.7 $s^{-1}$, 2.2±2.6 $s^{-1}$ and 1.2±1.7×10$^{-3}$ $s^{-1}$ (± standard
deviation), respectively. $NO_x$ (NO+$NO_2$) were by far the main contributors to the $R_{OH}^{total}$, $R_{NO_3}^{total}$
and $R_{O_3}^{total}$, with average values of 47, 99 and 99%, respectively. Isoprene dominated the OH and
$NO_3$ reactivity towards TVOCs ($R_{OH}^{TVOCs}$ and $R_{NO_3}^{TVOCs}$), accounting for 40% and 77%, respectively.
However, alkenes dominated the $O_3$ reactivity towards TVOCs ($R_{O_3}^{TVOCs}$), representing 66% of
$R_{O_3}^{total}$. $R_{OH}^{total}$, $R_{NO_3}^{total}$ and $R_{O_3}^{total}$ displayed a similar diurnal variation with the lowest during the
afternoon and the highest during rush hours, and the diurnal profile of $NO_x$ appears to be the major
driver for the diurnal profiles of $R_{OH}^{total}$, $R_{NO_3}^{total}$ and $R_{O_3}^{total}$. The calculated atmospheric oxidative
capacity (AOC) was up to 4.4×10$^8$ molecules cm$^{-3}$ s$^{-1}$ with campaign-averaged values of 3.1×10$^7$
molecules cm$^{-3}$ s$^{-1}$ dominated by OH radicals (2.9×10$^7$ molecule cm$^{-3}$ s$^{-1}$, 95%), $O_3$ (1.2×10$^6$
molecule cm$^{-3}$ s$^{-1}$, 4%) and $NO_3$ radicals (1.7×10$^5$ molecule cm$^{-3}$ s$^{-1}$, 1%). The reaction with OH
radicals was the dominant volatile organic compounds (VOCs) loss except for trans-2-butene, cis-
2-butene, trans-2-pentene, propylene, 1-butene, 1-pentene, 1-hexene, acetone and styrene, where
the reaction with $O_3$ was more important for their loss rates. Compared with anthropogenic
hydrocarbons, the oxidation by the $NO_3$ radical was more important for the nighttime integral of
isoprene loss rates. Overall, the present study may provide some useful suggestions for VOC
pollution control in the Xianghe and North China Plain. To better understanding the trace gas
reactivity and AOC, further studies, especially direct observations of the OH and $NO_3$ radical
concentrations and their reactivities, are required.
**Keywords:** VOCs; radical reactivity; atmospheric oxidation capacity; loss rate; North China Plain





## 1 Introduction

In the planetary boundary layer, overwhelming quantities of trace gases from both biogenic and anthropogenic origins (Guenther et al., 1995;Piccot et al., 1992;Chen et al., 2019;Li et al., 2019b;Wang et al., 2019b) are also a major part transformed by reactions with free radicals such as hydroxyl (OH) radicals, nitrate ($NO_3$) radicals, and ozone ($O_3$) on local to global scales (Atkinson and Arey, 2003;Heard and Pilling, 2003), with the dominant reaction depending on the time of day and specific trace gases. Ultimately, these processes lead to the formation of a series of important secondary pollutants, including tropospheric $O_3$ and secondary organic aerosol (SOA) (Goldstein and Galbally, 2007).

OH radicals control the daytime oxidation capacity of the atmosphere (Heard and Pilling, 2003), initiating and participating in many oxidation reaction processes. OH exhibits a high reactivity to many atmospheric trace gases, such as carbon monoxide (CO), nitrogen oxides ($NO_x=NO+NO_2$) and volatile organic compounds (VOCs) (Kovacs et al., 2003;Sadanaga et al., 2005). The total OH reactivity ($R_{OH}$) is the sum of the products of the concentrations and respective reaction rate coefficients for all gases that react with OH. The total OH reactivity is equivalent to the inverse of the lifetime of OH ($s^{-1}$) in the presence of those atmospheric constituents. $R_{OH}$ can be measured directly ($R_{OH}^{measured}$), modeled ($R_{OH}^{modeled}$) or calculated from individual trace gas measurements ($R_{OH}^{calculated}$). The online techniques used to determine $R_{OH}^{measured}$ include the flow tube with sliding injector method, a comparative rate method and a laser flash photolysis pump probe technique (Yang et al., 2016;Whalley et al., 2016;Lou et al., 2010). Based on these online methods, the values of $R_{OH}$ have been measured in urban, suburban, remote and forest areas during the last decade. The urban areas investigated included Nashville, USA (SOS) (Kovacs et al., 2003), New York, USA (PMTACS-NY2004) (Ren et al., 2006a), Mexico City, Mexico (MCMA-2003) (Shirley et al., 2006), Houston, USA (TRAMP2006) (Mao et al., 2010), Paris, France (MEGAPOLI) (Dolgorouky et al., 2012), London, UK (ClearfLo) (Whalley et al., 2016)and Beijing, China (Yang et al., 2017). The ranges of $R_{OH}^{measured}$ in these urban areas ranged from 1 $s^{-1}$ in clean air to 200 $s^{-1}$ in extremely polluted air in the atmospheric boundary layer, and $NO_x$, CO, formaldehyde and VOCs were the main contributors (Ferracci et al., 2018). The suburban areas investigated included Whiteface Mountain, USA (PMTACS-NY2002) (Ren et al., 2006b), Weybourne, UK (TORCH-2) (Lee et al.,



2010), Yufa, China (CAREBeijing-2006) (Lu et al., 2010), Backgarden, China (PRIDE-PRD) (Lou
et al., 2010), Jülich, Germany (HOx Comp) (Elshorbany et al., 2012), Ersa, Corsica (CARBOSOR-
ChArMeX) (Zannoni et al., 2017) and Heshan, China (Yang et al., 2017). The ranges of $R_{OH}^{measured}$
in these suburban areas ranged from 4.6 to 31.4 s$^{-1}$. $R_{OH}^{modeled}$ was modeled using zero-dimensional
box model based on the Regional Atmospheric Chemical Mechanism to compare them with the
measurements or calculations (Lou et al., 2010;Whalley et al., 2016;Ferracci et al., 2018;Yang et al.,
2017). $R_{OH}^{calculated}$ is the sum of the OH reactivities that are attributed to measured trace gases,
which is used extensively as a matrix to estimate the initial peroxyl radical (RO$_2$) formation rate
under optimum reaction conditions (Carter, 2012;Liu et al., 2008;Warneke, 2004). This matric does
not account for chain termination or propagation steps, nor does it properly capture differences in
VOC production of RO$_2$ during photolysis or reaction with other oxidants; however, this matric does
provide at least some useful approximation of the relative contribution of individual VOCs to
daytime photochemistry (Goldan et al., 2004;Benedict et al., 2019). The concentrations (in
molecules cm$^{-3}$) of trace gases and the reaction rate constants ($k_{trace\ gase+OH}$ in cm$^3$ molecule$^{-1}$ s$^{-1}$)
of the respective trace gases with the OH radical are the key factors for computing $R_{OH}^{calculated}$. In
general, the trace gases have been considered when calculating $R_{OH}^{calculated}$ include VOCs, CO,
NO$_x$ and SO$_2$. Reportedly, the contribution from the NO$_x$ exceeds 50% for the cities of Paris, Tokyo,
New York and Beijing, showing the large influence of traffic-related emissions on the $R_{OH}^{calculated}$
(Dolgorouky et al., 2012;Ren, 2003;Yang et al., 2017;Yoshino et al., 2006), but the contribution
from the VOCs reaches 50% in Mexico and Houston (Mao et al., 2010;Shirley et al., 2006).

As OH levels are vastly reduced during nighttime due to missing photolysis, the NO$_3$ formed by the
slow reaction of NO$_2$ +O$_3$→NO$_3$ +O$_2$ is the main initiator of nighttime oxidation chemistry in the
troposphere due to the lack of photolysis and its elevated mixing ratios at night (Asaf et al.,
2009;Geyer et al., 2001). NO$_3$ reacts significantly with unsaturated VOCs such as isoprene, certain
alkenes and aromatics via additions to a >C=C< double bond, which can initiate the formation of
peroxyl radicals (HO$_2$ and RO$_2$) and even of OH (Geyer et al., 2001). The high NO$_3$ mixing ratios
and the large reaction rate constants with several unsaturated VOCs result in NO$_3$ being the
dominant sink of many unsaturated VOCs during nighttime. The role of NO$_3$ as an oxidizing agent
may be assessed via its total reactivity $R_{NO_3}$ towards trace gases (or inverse lifetime s$^{-1}$). $R_{NO_3}$ is



an indication of nighttime oxidation rates of trace gases with direct impacts on $NO_x$ levels and
indirect impacts on heterogeneous $NO_x$ losses and $ClNO_2$ formation (Liebmann et al., 2017). As
frequently reported for $R_{OH}$, $R_{NO_3}$ can be measured online ($R_{NO_3}^{\text{measured}}$) or calculated from
summing loss rates for a set of reactive trace gases ($R_{NO_3}^{\text{calculated}}$). Previous work on $R_{NO_3}^{\text{measured}}$ has
revealed a strong diel variation. For instance, the $R_{NO_3}^{\text{measured}}$ obtained during the IBAIRN
campaign, which was carried out in the boreal forest of Finland, Hyytiälä, displayed a strong diel
variation with a campaign-averaged nighttime mean value of 0.11 s$^{-1}$ compared to a daytime value
of 0.04 s$^{-1}$ (Liebmann et al., 2018a), but varied from 0.005 to 0.1 s$^{-1}$ during nighttime and reached
values as high as 1.4 s$^{-1}$ in the daytime during NOTOMO (Liebmann et al., 2017).

Along with reactions with the OH and $NO_3$ radicals, trace gases are oxidized in the troposphere by
reactions with $O_3$. Although $O_3$ reacts significantly with alkenes, for most VOCs, the reaction rate
of $O_3$ is much slower than the reaction rate of OH and $NO_3$. However, $O_3$ is very important because
it is present at elevated mixing ratios in clean or contaminated atmospheres (Wang et al., 2013). The
rate constants for some reactions of $O_3$ with alkenes are even comparable to those with $NO_3$
(Atkinson and Arey, 2003). The total reaction frequency of $O_3$ with trace gases ($R_{O_3}$) can reflect the
role of $O_3$ as an oxidizing agent. Direct measurements of $R_{O_3}$ were not available until very recently
(Geyer, 2003); hence, the reactivity of $O_3$ has traditionally been calculated ($R_{O_3}^{\text{calculated}}$) by summing
the reactivities due to individual reactive trace gases. $R_{O_3}^{\text{calculated}}$ obtained during the BERLIOZ
campaign revealed that terpenes (20%), isoprene (20%), and other alkenes (60%) were the dominant
contributors during the night of 20 and 21 July but arose mainly (83%) from non-biogenic alkene
during the night of 4 and 5 August (Geyer, 2003).

As mentioned above, OH radical, $NO_3$ radical, and $O_3$-initiated reactions of trace gases with
different mechanisms result in different rate coefficients and thus different reactivities. Recently, a
few studies on $R_{OH}^{\text{measured}}$, $R_{OH}^{\text{modeled}}$ or $R_{OH}^{\text{calculated}}$ have been conducted in China (Lou et al.,
2010;Fuchs et al., 2017b;Yang et al., 2017;Lu et al., 2010;Williams et al., 2016;Lyu et al., 2019).
However, comprehensive evaluations of $R_{OH}^{\text{calculated}}$, $R_{NO_3}^{\text{calculated}}$ and $R_{O_3}^{\text{calculated}}$ are scarce. In
this study, $R_{OH}^{\text{calculated}}$, $R_{NO_3}^{\text{calculated}}$ and $R_{O_3}^{\text{calculated}}$ were conducted at a suburban site (Xianghe) in
the North China Plain during an intensive measurement campaign in the summer of 2018. By





combining OH and NO$_3$ concentrations determined using parameterization methods, the oxidation
capacities of OH, NO$_3$ and O$_3$ were compared to estimate their contributions to the atmospheric
oxidation capacity.

**2 Methodology**
**2.1 Site description**
The sampling site is located at the Xianghe Atmospheric Observatory (39.798 °N, 116.958 °E; 15
m above sea level), which is operated by the Institute of Atmospheric Physics (IAP)/Chinese
Academy of Sciences (CAS). The sampling site is a typical suburban site in the seriously polluted
Beijing-Tianjin-Hebei large urban region, which is approximately 50 km southeast of Beijing, 75
km northwest of Tianjin, and 35 km northeast of Langfang in the Hebei Province. The sampling site
is approximately 4 km west of the downtown center and is surrounded by residential areas and
agricultural land (see Figure 1).

**2.2 Experimental method**
**Criteria pollutants** O$_3$ was measured using a UV photometric O$_3$ analyzer (Model 49C/I, Thermo-
Fisher Scientific, United States) with the lowest detection limit of 2.0 ppb, precision of ±1.0 ppb,
zero drift of less than 1.0 ppb (24 h)$^{-1}$, span drift of less than 1% full scale per month, and response
time of 10 s. NO$_x$ was measured using a chemiluminescence NO$_x$ Analyzer (Model 42C/I) with the
lowest detection limit of 0.4 ppb, precision of ±0.4 ppb, zero drift of less than 0.4 ppb (24 h)$^{-1}$, span
drift of less than 1% per 24 h, and response time of 40 s. NO$_y$ was measured using a
chemiluminescence NO-DIF-NOy Analyzer (Model 42C/I) with the lowest detection limit of 50 ppt,
span drift of less than 1% per 24 h, and response time of 60 s. SO$_2$ was measured using a pulsed
fluorescence SO$_2$ analyzer (Model 43C/I) with the lowest detection limit of 0.5 ppb, precision of 1%
of reading or 1 ppb, zero drift of less than 1 ppb (24 h)$^{-1}$, span drift of less than 0.5% full scale per
24 h, and response time of less than 20 s. CO was measured with a nondispersive infrared analyzer
(Model 48I) with the lowest detection limit of 0.4 ppm, zero drift of less than 0.1 ppb (24 h)$^{-1}$, span
drift of less than 0.1% full scale per 24 h, and response time of less than 60 s. The PM$_{2.5}$ was
measured by RP1400a TEOM micro-oscillation balance ambient particulate monitor with a
resolution of 0.1 μg m$^{-3}$, a detection limit of 0.06 μg m$^{-3}$ (1-h average), and precisions of ±1.5 μg m$^{-}$



[3] (1-h average) and ±0.5 µg m[-3] (24-h average). The entire system was heated to 50 °C, thus, loss of
semi-volatile compounds thereby cannot be avoided. Depending on the ammonium nitrate levels
and ambient temperatures, up to 25 % lower mass concentrations were found for select daily means
compared with gravimetric filter measurements. The sampling methods and instrument protocols as
well as quality assurance/quality control (QA/QC) procedures for air quality monitoring are
described in detail in the Chinese National Environmental Protection Standard, Automated Methods
for Ambient Air Quality Monitoring (HJ/T 193–2005; State Environmental Protection
Administration of China, 2006). The measurement techniques are the same as those used in (Wang
et al., 2014b;Xin et al., 2010).

**Volatile organic compounds** Ambient VOC samples were collected and analyzed continuously and
automatically with a time resolution of 1 h using a custom-built gas chromatography-mass
spectrometry/flame ionization detector (GC-MS/FID). The availability of this system for VOCs
measurement are well verified and it has been used in several large field campaign (Chen et al.,
2014;Yuan et al., 2013;Wu et al., 2016). The online GC-MS/FID system consisted of three major
components: a cryogen-free cooling device for creating ultra-low temperatures (TH300, Wuhan
Tianhong Environmental protection industry co., LTD, Wuhan, China), a sampling and
preconcentration system for VOC collection and enrichment, and a gas chromatography (GC,
7820A, Agilent Technologies, Santa Clara, CA, USA) equipped with an MS and an FID (5977E,
Agilent Technology, Santa Clara, CA, USA) for VOC separation and detection (Wang et al., 2014a).
A complete analysis cycle for ambient VOC measurements by the online GC-MS/FID system
includes five stages: preparation, sampling and preconcentration, injection/GC analysis, idle/GC
analysis, and back purge/GC analysis. Briefly, moisture and $CO_2$ are removed before VOC analysis.
Most C2-C5 hydrocarbons were separated on a PLOT-$Al_2O_3$ column (15 m×0.32 mm ID×3 µm,
J&W Scientific, USA), and measured by the FID channel. Other compounds were separated on a
semi-polar column (DB624, 60 m×0.25 mm ID×1.4 µm, J&W Scientific, USA) and quantified using
a quadrupole MS detector.

The compounds analyzed were subjected to rigorous quality assurance and quality control
procedures (QA/QC). The VOCs detected by FID were quantified by the external standard method,



and the components detected by MS were quantified by the internal standard method. Four
compounds, i.e., bromochloromethane, 1,4-difluorobenzene, chlorobenzene-d5, and
bromofluorobenzene, were used as internal standards. Specifically, the system was calibrated at
multiple concentrations in the range of 0.8-8 ppb by two gas standards, i.e., a mixture of 57 PAMS
(provided by Spectra Gases Inc., USA), and a mixture of oxygenated VOCs (OVOCs) and
halocarbons (provided by Spectra Gases Inc., USA). $R^2$ values for the calibration curves ranged
from 0.941(n-Dodecane) to 1.000 for VOCs, indicating that integral areas of the peaks were
proportional to concentrations of target compounds. The method detection limit (MDL) of the online
GC-FID/MS system for all measured compounds ranged from 0.003 to 0.092 ppb. The measurement
relative standard deviation (RSD) for measured compounds ranged from 2.1% to 14.9% (Yang et
al., 2019). To check the stability of the instrument, routine calibration was performed periodically
by using a calibration gas with a mixing ratio of 2 ppb consisting of 56 kinds of VOC components.
The variations between the measured and nominal concentrations of the periodic calibration were
within 10%. The signal variations of each targeted compound due to system instability were
corrected by the signal of CFC-113 (1,1,2-trichloro-1,2,2-trifluoroethane) due to its long
atmospheric lifetime and stable anthropogenic emissions (Yuan et al., 2013;Chen et al., 2014).
Detailed instrumental and operational parameters are described in our previous study (Yang et al.,

226    2019).


**Photolysis frequency** The photolysis frequencies, $J_{O^1D}$, $J_{NO_2}$ and $J_{NO_3}$, in the atmosphere are
measured by the PFS-100 Photolysis Spectrometer (Juguang Technology (hangzhou) Co., Ltd,
Hangzhou, China). The photolysis rate is calculated by integrating the actinic flux with the known
absorption cross section $\sigma(\lambda)$ and quantum yield $\varphi(\lambda)$. The actinic flux is spherically integrated
photon radiance of the solar radiation in the atmosphere. The spectrometer obtains spectral
information in a certain wavelength range, which mainly uses quartz receiver to collect solar
radiation from all directions, and convert it into the actinic flux $F_\lambda$. $\sigma(\lambda)$ is the absorption cross
section of the species that absorbs in certain wavelength range and $\varphi(\lambda)$ is quantum yield of the
photodissociation reaction product; these two coefficients have been measured by experiments and
can be directly looked up and used.



**Aerosol surface area density** Particle surface concentrations in the range of 10-9486.8 nm
(mobility diameter, dm) were measured using WPS (Model 1000XP Wide Range Particle
Spectrometer, MSP Corporation, USA). The instruments provided continuous measurements during
the whole observation except for the maintenance of instruments and power outages. Ambient air
was drawn into a stainless steel tube with a length of 3.0 m and an inner diameter of 0.5 inch in via
a steel dust seal. From this tube, ambient air from the split flow was drawn through a conductive
silicone tubing with a 1/4 inch in inner diameter via a stainless tube with a length of 0.5 cm into the
WPS at rates of 1.0 L·min$^{-1}$. The overall RH was maintained below 50% by a Nafion tube to avoid
water condensation within the inlet systems.

**Meteorology parameters** In addition, the meteorological parameters, including wind speed, wind
direction, temperature and relative humidity were obtained from the National Meteorological
Information Center (http://data.cma.cn/).

**2.3 Speciated radical reactivity**
Radical reactivity is a measure of the strength of the sinks for the radical (Sonderfeld et al.,
2016;Fuchs et al., 2017a;Fuchs et al., 2017b;Tan et al., 2019). Total radical reactivity (s$^{-1}$) is defined
as the total radical loss rate, which is the inverse of its lifetime with respect to a radical in the
atmosphere (Di Carlo et al., 2004;Mao et al., 2009;Mao et al., 2010;Liebmann et al., 2017;Liebmann
et al., 2018b). High radical reactivity values correspond to short lifetimes and long-lived species
have low reactivities.
For total OH reactivity
$$R_{OH} = \sum K_{OH+VOC_i}[VOC_i] + K_{OH+CO}[CO] + K_{OH+NO}[NO] + K_{OH+NO_2}[NO_2]$$
$$+K_{OH+SO_2}[SO_2] + K_{OH+O_3}[O_3] + \cdots \qquad (1)$$
For total NO$_3$ reactivity
$$R_{NO_3} = \sum K_{NO_3+VOC_i}[VOC_i] + K_{NO_3+NO}[NO] + K_{NO_3+NO_2}[NO_2]$$
$$+K_{NO_3+SO_2}[SO_2] + \cdots \qquad (2)$$
For total O$_3$ reactivity
$$R_{O_3} = \sum K_{O_3+VOC_i}[VOC_i] + K_{O_3+NO}[NO] + K_{O_3+NO_2}[NO_2] + \cdots \qquad (3)$$





In the above equations, the pseudo first order rate coefficients (in $cm^3$ molecule$^{-1}$ s$^{-1}$) for OH-
$VOC_i$ ($K_{OH+VOC_i}$), OH-CO ($K_{OH+CO}$), OH-NO ($K_{OH+NO}$), OH-NO$_2$ ($K_{OH+NO_2}$), OH-SO$_2$ ($K_{OH+SO_2}$),
OH-O$_3$ ($K_{OH+O_3}$), NO$_3$-$VOC_i$ ($K_{NO_3+VOC_i}$), NO$_3$-NO ($K_{NO_3+NO}$), NO$_3$-NO$_2$ ($K_{NO_3+NO_2}$), NO$_3$-SO$_2$
($K_{NO_3+SO_2}$), O$_3$-$VOC_i$ ($K_{O_3+VOC_i}$), O$_3$-NO ($K_{O_3+NO}$) and O$_3$-NO$_2$ ($K_{O_3+NO_2}$) were based on
recommended values from the International Union of Pure and Applied Chemistry (IUPAC)
Subcommittee for Gas Kinetic Data Evaluation (http://iupac.pole-ether.fr, last accessed: 25 July
2018), the JPL-NASA Evaluation of Chemical Kinetics and Photochemical Data for Use in
Atmospheric Studies (Atkinson et al., 2004;Atkinson et al., 2006b) and the Master Chemical
Mechanism, MCM v3.2 (Ferracci et al., 2018), via the website: http://mcm.leeds.ac.uk/MCM (last
accessed: 25 July 2019); $[VOC_i]$, $[CO]$, $[NO]$, $[NO_2]$, $[SO_2]$ and $[O_3]$ are their concentrations
(in molecules $cm^{-3}$), respectively.

**2.4 Atmospheric oxidation capacity**
The term "oxidation capacity" of an oxidant $X$ (= NO$_3$, OH and O$_3$) is defined as the sum of the
respective oxidation rates of the molecules $Y_i$ (Geyer et al., 2001).

$$AOC = \sum_{i=1} k_{Y_i-X}[Y_i]\,[X] = \sum_{i=1} R_X^{Y_i}\,[X] \quad (4)$$

Here, $[Y_i]$ and $[X]$ are mixing ratios of molecule $Y_i$ and oxidant $X$, respectively. $k_{Y_i-X}$ is the
rate constant of the molecule $Y_i$ with oxidant $X$. $R_X^{Y_i}$ is the oxidant $X$ reactivity of molecules $Y_i$.

Simultaneous measurements of OH and NO$_3$ are not available in this study. The OH radical
concentration (in molecule $cm^{-3}$) can be estimated using the expression for the dependence of the
OH concentration on solar UV and NO$_2$ suggested by (Ehhalt and Rohrer, 2000) and verified by
(Alicke, 2002):

$$[OH] = a \times (J_{O^1D})^\alpha \times (J_{NO_2})^\beta \times \frac{b \times [NO_2] + 1}{c \times [NO_2]^2 + d \times [NO_2] + 1} \quad (5)$$

Here, $J_{O^1D}$ and $J_{NO_2}$ are the measured photolysis frequencies (s$^{-1}$) of O$_3$ and NO$_2$, respectively.
The values of $\alpha = 0.83$, $\beta = 0.19$, a=4.1×10$^9$, b=140, c=0.41 and d=1.7 are obtained from
measurement data with NO$_x$ concentrations > 1 ppb during the POPCORN campaign at a rural site
in Germany.





The $NO_3$ concentration (in molecule $cm^{-3}$) could be determined based on the steady-state
assumption of the $NO_3$ concentration in the atmosphere (Yuan et al., 2013):
$$[NO_3] = \frac{k_{NO_2+O_3} \times [NO_2] \times [O_3]}{J_{NO_3} + k_{NO+NO_3} \times [NO] + R_{NO_3}^{VOCs} + K_{eq} \times \frac{\gamma \bar{c} A}{4} \times [NO_2]} \quad (6)$$

Here, $J_{NO_3}$ is the measured photolysis frequency $(s^{-1})$ of $NO_3$. $R_{NO_3}^{VOCs}$ $(s^{-1})$ is the VOC
reactivity to $NO_3$. The rate coefficients for $NO_2$-$O_3$ $(k_{NO_2+O_3})$ and NO-$NO_3$ $(k_{NO+NO_3})$ were taken
from the JPL-NASA Evaluation of Chemical Kinetics and Photochemical Data for Use in
Atmospheric Studies (Atkinson et al., 2004) and are $3.5 \times 10^{-17}$ and $2.6 \times 10^{-11}$ $cm^3$ $molecule^{-1}$ $s^{-1}$,
respectively. $K_{eq}$ $(=3.26 \times 10^{-11}$ $cm^3$ $molecule^{-1}$ $s^{-1}$ at 298K) is the equilibrium constant for $NO_3$
+$NO_2$ +M⇌$N_2O_5$ +M (Atkinson et al., 1986). $\gamma$ (=0.022) is the dimensionless uptake coefficient
obtained in North China Plain (Tham et al., 2018). $\bar{c}$ is the mean molecular velocity of $N_2O_5$
(26233cm $s^{-1}$ at 298 K). A is the aerosol surface area density $(cm^2$ $cm^{-3})$. However, simultaneous
measurement of aerosol surface area density with VOC is not available in this study. We calculated
the aerosol surface area density at the site on a linear fitting equation (aerosol surface area density
=$415.32 \times [PM_{2.5}]+6511.6$ $R^2$=0.7846 p<0.001) between aerosol surface area density and $PM_{2.5}$
measured from 1 to 22 November 2018 , as showed in Figure S1. Figure S2 shows the time series
of calculated aerosol surface area density. The campaign-averaged values of aerosol surface area
density was $2.35 \times 10^{-6}$ $cm^2$ $cm^{-3}$ with a range of $7.21 \times 10^{-7}$- $5.48 \times 10^{-6}$ $cm^2$ $cm^{-3}$.

**3 Results and discussion**
**3.1 Overview of measurements**
For the data evaluation, all measurements were averaged over 1-hour time intervals. The measured
concentrations of major pollutants and meteorological parameters at Xianghe are depicted in Figure
2, while the mean diurnal profiles are shown in Figure S3. During the campaign, sunny weather
conditions prevailed with temperatures ranging from 25℃ to 31℃ during the daytime. The ambient



temperature was comparable with those measured in Beijing (02 Jul-19 Jul 2014), Shanghai (21
Aug-02 Sep 2016), and Chongqing (27 Aug-04 Sep 2015), but higher than that in Guangzhou (23-
31 Oct 2015) (Tan et al., 2019). Wind data suggested that the prevailing wind was from the eastern
sampling site with a mean wind speed of 1.0 m s$^{-1}$ ranging from 0.3 m s$^{-1}$ to 1.4 m s$^{-1}$, and the
average relative humidity was 85%, reaching up to 96% during the night (Figure 2). For the
campaign, $NO_y$ showed a morning peak with a maximum of 228.8 ppb at 9:00 h and an afternoon
dip with a minimum of 26.1 ppb at 16:00 h (Figure S3a). Campaign-averaged data maximum and
minimum $SO_2$ mixing ratios of 3.6 ppb at approximately 14:00 h and 2.3 ppb during nighttime were
obtained (Figure S3c). For $J_{O^1D}$, $J_{NO_2}$ and $J_{NO_3}$, a similar maximum at ~14:00 h was observed,
with maximum values of 2.1 ×10$^{-5}$ s$^{-1}$, 5.3 ×10$^{-3}$ s$^{-1}$ and 1.3 ×10$^{-1}$ s$^{-1}$, respectively (Figure S3g-Figure
S3i). The maximum of $J_{O^1D}$ at this site was comparable with that in Shanghai and Chongqing but
higher than that in Guangzhou and lower than that in Beijing (Tan et al., 2019;Wang et al., 2019a).
The observed mean daily maxima of $J_{NO_2}$ at this site were higher than those observed in the eastern
Mediterranean (Gerasopoulos et al., 2012) but lower than that in Beijing (Wang et al., 2019a).

The diurnal maximum $O_3$ concentration was 72 ppb at this site (Figure S3d), which was in line with
that observed in Beijing (72 ppb) but higher than that measured in Guangzhou (65 ppb) and
Chongqing (56 ppb) and lower than that observed in Shanghai (80 ppb) (Tan et al., 2019). The $O_3$
precursors, CO, $NO_x$, and VOCs, are shown in Figure 2 and Figure S3. As expected, with the
accumulation of CO, $NO_x$, and VOCs, the $O_3$ concentration gradually increases, and the
concentration of VOCs gradually decreases as the photochemical reaction progresses (Kansal,
2009;Song et al., 2018). CO and $NO_x$ showed a similar diurnal profile with a maximum during the
rush hour and a minimum in the afternoon (Figure S3b and Figure S3e), suggesting that both CO
and $NO_x$ originated from the same source (enhanced traffic emission), and/or were manipulated by
the same factor (e.g., poor dilution conditions). During the campaign, the average mixing ratio of
total VOCs was 25.3 ppb, with the highest contributions from alkanes (13.2 ppb, 51.4%), followed
by OVOCs (4.9 ppb, 19.8%), aromatics (4.3 ppb, 16.7%) and alkenes (3.0 ppb, 12.1%). The top 10
species (Figure 3a), in terms of emissions, consisted of propane (3.7 ppb), acetone (3.2 ppb), ethane
(3.2 ppb), n-butane (1.9 ppb), m/p-Xylene (1.6 ppb), iso-pentane (1.3 ppb), ethylene (1.3 ppb), iso-
butane (1.1 ppb), isoprene (1.0 ppb) and n-pentane (0.7 ppb), accounting for a total of 75.1% of the





TVOC concentration. As typical tracers of vehicle-related emissions, propane, ethane, ethene,
butanes and pentanes were present in high concentrations, suggesting that vehicle-related emissions
were likely to be the dominant source of VOCs at this site. In addition, the shape of diurnal variations
of TVOCs backed the presence of vehicle-related emissions, which presented relatively higher
mixing ratios during the early morning and from evening to midnight, which may be related to
enhanced traffic emissions during rush hours and poor dilution conditions (Yuan et al., 2009;He et
al., 2019;Tan et al., 2019). On the other hand, the mixing ratios of TVOCs began to decrease at
10:00 h and maintained a broad trough during daytime hours probably due to the increased
photochemical removal processes favoring the destruction of VOCs, elevated planetary boundary
layer (PBL) advancing the dispersion of VOCs and/or less VOC emissions reducing levels of VOCs
(He et al., 2019;Zheng et al., 2018). In contrast, the OVOC concentrations (not shown) increased
from a minimum near sunrise and a maximum in the late afternoon, reflecting the accumulation of
OVOCs during the photochemically active period of the day and illustrating the time profile of
formation for a secondary species (Yuan et al., 2012).

**3.2 Reactivity of hydroxy radical, nitrate radical and ozone**
In this study, $R_{OH}^{calculated}$, $R_{NO_3}^{calculated}$ and $R_{O_3}^{calculated}$ were comprehensively conducted. All the
reactivity values discussed in this study were calculated rather than observed. Figure 4 shows the
time series of calculated $R_{OH}^{calculated}$, $R_{NO_3}^{calculated}$ and $R_{O_3}^{calculated}$. The contributions of different
atmospheric compounds to $R_{OH}$ are presented in Figure 5. The mean diurnal variations in
$R_{OH}^{calculated}$, $R_{NO_3}^{calculated}$ and $R_{O_3}^{calculated}$ are shown in Figure 6 and Figure 7. The frequency
distributions of $R_{OH}^{calculated}$, $R_{NO_3}^{calculated}$ and $R_{O_3}^{calculated}$ are depicted in Figure S4-Figure S9.

**3.2.1 OH reactivity ($R_{OH}$)**
The $R_{OH}$ of trace gases was categorized into $SO_2$, CO, $NO_x$ (sum of NO and $NO_2$) and TVOCs,
which were grouped into alkanes, alkenes, aromatics, OVOCs and isoprene (Table S1 lists the VOCs
included in each group), as shown in Figure 4a and 4b. The total $R_{OH}$, $R_{OH}^{total}$, was between 8.5 and
68.1 $s^{-1}$, with an average of 25.6±9.7 $s^{-1}$ (± standard deviation). Statistically, the average $R_{OH}^{total}$
was much higher than those determined in Beijing (16.4 $s^{-1}$/20±11 $s^{-1}$) (Tan et al., 2019;Yang et al.,
2017), Shanghai (13.5 $s^{-1}$) (Tan et al., 2019), Chongqing (17.8 $s^{-1}$) (Tan et al., 2019), Jinan (19.4±2.1



s$^{-1}$) (Lyu et al., 2019), Wangdu (10-20 s$^{-1}$) (Fuchs et al., 2017b), Houston (9-22 s$^{-1}$) (Mao et al.,

2010), London (18.1 s$^{-1}$) (Whalley et al., 2016) and Nashville (11.3 ± 4.8 s$^{-1}$) (Kovacs et al., 2003)

but was comparable or lower than those in Guangzhou (22.7 s$^{-1}$) (Tan et al., 2019), Heshan (31±20

s$^{-1}$) (Yang et al., 2017), Backgarden (mean maximum value of 50 s$^{-1}$) (Lou et al., 2010) and New

York (25 s$^{-1}$) (Ren et al., 2006b). The $R_{OH}^{\text{total}}$ was mainly contributed by NO$_x$ (12.0±7.1 s$^{-1}$, 47%),

followed by CO (7.2±2.6 s$^{-1}$, 28%) and TVOCs (6.2±4.6 s$^{-1}$, 24%) and to a lesser extent by SO$_2$

and O$_3$ (0.2±0.1 s$^{-1}$, 1%), indicating the strong influence of anthropogenic emissions in Xianghe.

The majority of $R_{OH}^{\text{total}}$ values were below 20 s$^{-1}$, as seen in the frequency distribution, which was

dominated by the sum of low $R_{OH}$ contributions and less by single compounds with high $R_{OH}$

(Figure S4a-S4e), highlighting the necessity of considering a large number of species to obtain a

complete picture of $R_{OH}^{\text{total}}$. Specifically, the cumulative frequency distribution (Figure S5a) clearly

showed that the $R_{OH}^{\text{total}}$ at values >40 s$^{-1}$ was dominated entirely by $R_{OH}^{\text{NO}_x}$, and the $R_{OH}^{\text{total}}$ at values

between 20-40 s$^{-1}$ was nearly dominated by $R_{OH}^{\text{NO}_x}$ and $R_{OH}^{\text{VOCs}}$.

Figure 5 presents the contributions from different atmospheric constituents, including CO, NOx, the

sum of nonmethane hydrocarbons (NMHC), OVOCs and the sum of biogenic VOC (BVOCs), to

the $R_{OH}$ for 12 different urban atmospheric measurements around the world and different periods

of the year. In total, the contributions from the inorganic species (CO and NO$_x$) exceeded 50% for

Xianghe, Beijing (Tan et al., 2019), Shanghai (Tan et al., 2019), Guangzhou (Tan et al., 2019),

Chongqing (Tan et al., 2019), Heshan (Yang et al., 2017), Paris (Dolgorouky et al., 2012), Tokyo

(Yoshino et al., 2006) and New York (Mao et al., 2010), showing the large influence of traffic-

related emissions on $R_{OH}^{\text{total}}$. The contributions from CO and NO$_x$ were very similar in Xianghe

(75%) and Paris (76%), although different seasons are considered, suggesting a possible influence

from traffic emissions (Dolgorouky et al., 2012). In contrast, the NMHCs contributed between 12%

and 28% to the $R_{OH}^{\text{total}}$ in these cities. However, the contributions from CO and NO$_x$ in Mexico City

and Houston (Mao et al., 2010) were only 37% and 29%, respectively, but the contributions from

the NMHCs reached 51% and 46%, respectively. This was accounted for by (1) Mexico City sharing

high NMHC due to higher biomass fuel being burned (de Gouw et al., 2006) and (2) higher

contributions from aromatics due to high industrial solvent emissions in Houston (Leuchner and

Rappenglück, 2010). In conclusion, the $R_{OH}$ was a typical fingerprint of anthropogenic emissions





(traffic-related emissions or industrial emissions), although comparing $R_{OH}$ in different places is a
limited exercise because they are by nature a point measurement that can vary inside the same city
depending on the geographic location of the measurement and the season (Dolgorouky et al., 2012).

The $R_{OH}$ of TVOCs, $R_{OH}^{\text{TVOCs}}$, was 6.2±4.6 s$^{-1}$, which was much lower than those in Beijing (11.2
s$^{-1}$) and Heshan (18.3 s$^{-1}$) (Yang et al., 2017) due to the higher content of reactive hydrocarbons
(e.g., alkenes and aromatics) in Beijing and Heshan and the unmeasured species (e.g., HCHO and
acetaldehyde) in this study. Isoprene (2.5±3.7 s$^{-1}$, 40%) dominated over aromatics (1.5±1.7 s$^{-1}$,
24%), alkenes (0.9±0.8 s$^{-1}$, 14%), OVOCs (0.7±0.8 s$^{-1}$, 12%) and alkanes (0.7±0.5 s$^{-1}$, 10%) in the
$R_{OH}^{\text{TVOCs}}$. The majority of $R_{OH}^{\text{VOCs}}$ values were below 2 s$^{-1}$ (Figure S6a-S6e). The cumulative
frequency distribution showed that $R_{OH}^{\text{TVOCs}}$ at values of >15 s$^{-1}$ was dominated entirely by
$R_{OH}^{\text{isoprene}}$, the $R_{OH}^{\text{TVOCs}}$ at values between 10-15 s$^{-1}$ was dominated by $R_{OH}^{\text{isoprene}}$ and $R_{OH}^{\text{aromatics}}$,
and the $R_{OH}^{\text{TVOCs}}$ at values between 5-10 s$^{-1}$ was nearly dominated by $R_{OH}^{\text{isoprene}}$, $R_{OH}^{\text{aromatics}}$ and
$R_{OH}^{\text{OVOCs}}$ (Figure S7). Alkanes accounted for >50% of the mixing ratio of VOCs, but only 10% of
the $R_{OH}^{\text{TVOCs}}$. In contrast, aromatics, alkenes and OVOCs accounted for 44.6% of the mixing ratio
of VOCs, providing 50% of the $R_{OH}^{\text{TVOCs}}$. Significantly, isoprene accounted for only 4% of the
mixing ratio of VOCs but provided 40% of the $R_{OH}^{\text{TVOCs}}$. This was explained by (1) the relatively
low concentration of aromatics, alkenes and OVOCs measured during the campaign, (2) the
relatively high concentration of isoprene and (3) the generally large isoprene reaction rate
coefficients with OH (101×10$^{-12}$ cm$^3$ molecule$^{-1}$ s$^{-1}$) (Atkinson et al., 2006a). The $R_{OH}$ from
isoprene in this study was much higher than those in Guangzhou (0.4 s$^{-1}$), Beijing (1 s$^{-1}$), Chongqing
(1 s$^{-1}$) (Tan et al., 2019) and Heshan (0.9 s$^{-1}$) (Yang et al., 2017). The $R_{OH}$ from alkenes, aromatics
and OVOCs were dominated by m/p-xylene (0.8±0.9 s$^{-1}$), ethylene (0.3±0.2 s$^{-1}$), hexanal (0.2±0.4
s$^{-1}$), o-xylene (0.2±0.3 s$^{-1}$), propylene (0.2±0.2 s$^{-1}$) and styrene (0.2±0.6 s$^{-1}$) (Figure 3b). In total,
these 6 species contributed 31% of the $R_{OH}^{\text{TVOCs}}$ from 17% of the TVOC emissions.

The mean diurnal profiles of the $R_{OH}$ of trace gases and VOC groups are presented in Figure 6a-
6e and Figure 7a-7e, respectively. In general, the $R_{OH}^{\text{total}}$ was the lowest in the afternoon and the
highest during rush hours, reaching a maximum of 31 s$^{-1}$ during the morning rush hour and a night-
time peak of 28 s$^{-1}$ (Figure 6a). Most campaigns have also reported a slightly higher $R_{OH}$ in the





morning traffic rush hour, which can be explained by higher levels of reactive gases such as NO and
VOCs due to heavy traffic, as well as slower reactions (Yang et al., 2016;Dolgorouky et al.,
2012;Fuchs et al., 2017b;Mao et al., 2010;Ren, 2003;Ren et al., 2006a;Williams et al., 2016). A
similar diurnal profile was also observed for contributions from $NO_x$, CO, alkane, alkene and
aromatic species, which are typically connected to emissions from anthropogenic activities. The
shape of the $R_{OH}^{total}$ diurnal pattern was slightly shifted to the $R_{OH}^{NO_x}$, strengthening the idea that the
local pollution in Xianghe was possibly impacted by traffic emissions. However, a different diurnal
behavior to that of the above species was observed for OVOCs (Figure 7d) and isoprene (Figure 7e),
which is emitted by plants or photochemical production. The $R_{OH}$ from OVOCs increased by a
factor of approximately 1.6 from nighttime to daytime, suggesting that during the daytime, dilution
or chemical removal had a weaker influence on the observed OVOCs than fresh production by
photochemistry. The opposite diurnal variation was reported in Wangdu, which showed a weak
diurnal variation with a decrease by a factor of approximately 2 from the morning to the evening
(Fuchs et al., 2017b). Biogenic isoprene is dependent on temperature and light intensity (Guenther
et al., 1993;Pacifico et al., 2009;Saunier et al., 2017;Ding et al., 2014) and anthropogenic isoprene
is predominantly emitted by road traffic (Derwent et al., 1995;Ye et al., 1997); hence, the $R_{OH}$ from
isoprene increased during the daytime, with a morning peak of 3 s$^{-1}$ at 10:00 h and a night-time peak
of 7 s$^{-1}$ at 19:00 h. Many rainforest campaigns have also reported a significant diurnal pattern with
higher $R_{OH}$ from isoprene and OVOCs at noontime or reached a maximum at the beginning of the
night (Edwards et al., 2013;Sinha et al., 2008;Yang et al., 2016;Zannoni et al., 2016;Bsaibes et al.,
2019;Kaiser et al., 2016;Ramasamy et al., 2016). Notably, the large amplitude of standard deviation
bars highlighted the large diel variability.

**3.2.2 NO₃ reactivity ($R_{NO_3}$)**
The $R_{NO_3}$ of trace gases was categorized into $SO_2$, $NO_x$ and TVOCs which were grouped into
alkanes, alkenes, aromatics, OVOCs and isoprene (Table S1 lists the VOCs included in each group),
as shown in Figure 4c and 4d. Campaign-averaged values of $R_{NO_3}^{total}$ were 2.2±2.6 s$^{-1}$ (± standard
deviation) ranging from as low as 0.7 s$^{-1}$ to as high as 27.5 s$^{-1}$. The average $R_{NO_3}^{total}$ was much
higher than those determined during the IBAIRN campaign (Influence of Biosphere-Atmosphere
Interactions on the Reactive Nitrogen budget) (Liebmann et al., 2018a) and at a rural mountain site


(988 m a.s.l.) in southern Germany in 2017 (Liebmann et al., 2018b) due to the higher contributions
from $NO_x$ in this study. We noted that $NO_x$ was by far the main contributors to the $R_{NO_3}^{total}$,
representing 99% of the $R_{NO_3}^{total}$ on average. NO exhibited the most prominent contribution to
the $R_{NO_3}^{total}$ and represented an average of 78% of the $R_{NO_3}^{total}$. In comparison to NO, $NO_2$ had a
maximum contribution during night-time and represented, on average, 27% of the $R_{NO_3}^{total}$.
The $R_{NO_3}$ of VOCs and $SO_2$ was very minor, with no more than 1% of the $R_{NO_3}^{total}$ over the whole
campaign. The majority of $R_{NO_3}^{total}$ values were below 5 s$^{-1}$, but below 5×10$^{-2}$ s$^{-1}$, 5 s$^{-1}$, 5×10$^{-9}$ s$^{-1}$
for $R_{NO_3}^{TVOCs}$, $R_{NO_3}^{NO_x}$ and $R_{NO_3}^{SO_2}$, respectively, as seen in the frequency distribution (Figure S4f-S4i).
The cumulative frequency distribution clearly showed that the $R_{NO_3}^{total}$ at low and high values was
entirely dominated by $R_{NO_3}^{TVOCs}$ and $R_{NO_3}^{NO_x}$, respectively (Figure S5b).

The $R_{NO_3}$ of TVOCs, $R_{NO_3}^{TVOCs}$, was 2.2±2.7×10$^{-2}$ s$^{-1}$ on average with a minimum of 1.0×10$^{-3}$ s$^{-1}$
and a maximum of 0.2 s$^{-1}$. The largest fraction of attributed $R_{NO_3}^{TVOCs}$ was provided by isoprene
(77%), followed by aromatics (15%), alkene (7%) and OVOCs (1%). The measured alkanes played
virtually no role for $R_{NO_3}^{TVOCs}$, although they accounted for more than 50% of the mixing ratio of
VOCs. This can be largely explain by the fact that the reaction rate coefficients of isoprene,
aromatics, alkene and OVOCs with $NO_3$ are 1-5 orders of magnitude higher than the alkane reaction
rate coefficients with $NO_3$ (Atkinson et al., 2006b;Atkinson and Arey, 2003;Yuan et al.,
2013;Ferracci et al., 2018;Jenkin et al., 2015). The majority of $R_{NO_3}^{alkanes}$, $R_{NO_3}^{alkenes}$, $R_{NO_3}^{aromatics}$,
$R_{NO_3}^{OVOCs}$ and $R_{NO_3}^{isoprene}$ are below 5.0×10$^{-5}$ s$^{-1}$, 3.0×10$^{-3}$ s$^{-1}$, 1.0×10$^{-2}$ s$^{-1}$, 1.0×10$^{-3}$ s$^{-1}$ and 1.0×10$^{-3}$
s$^{-1}$, respectively (Figure S6f-S6j). The cumulative frequency distribution showed that $R_{NO_3}^{TVOCs}$ at
values of > 0.1 s$^{-1}$ was entirely dominated by $R_{NO_3}^{isoprene}$, the $R_{NO_3}^{TVOCs}$ at values between 0.01-0.1 s$^{-1}$
was dominated by $R_{NO_3}^{isoprene}$ and $R_{NO_3}^{aromatics}$, and the $R_{NO_3}^{TVOCs}$ at values of <1.0×10$^{-5}$ s$^{-1}$ was
entirely dominated by $R_{NO_3}^{alkanes}$ (Figure S8). The top seven species in terms of $R_{NO_3}^{TVOCs}$ consisted
of styrene, cis-2-butene, trans-2-butene, cis-2-pentene, hexanal, propylene and 1,3-butadiene
(Figure 3c). These species contributed only 4% to VOC emissions but accounted for 50% of the
$R_{NO_3}^{TVOCs}$.



$R_{NO_3}^{\text{total}}$ displayed a weak diel variation with a campaign-averaged morning peak value of $4.0\,\text{s}^{-1}$ at
6:00-7:00 h (Figure 6f). The diurnal profile of $R_{NO_3}^{NO_x}$ (Figure 6h) appears to be the major driver for
the diurnal profile of $R_{NO_3}^{\text{total}}$. The morning peak value of $R_{NO_3}^{\text{total}}$ could be explained by the
accumulation of $NO_x$ due to traffic emissions that are released into the shallow nocturnal boundary
layer during the morning rush hours. In contrast, the average diurnal profile of $R_{NO_3}^{\text{TVOCs}}$ (Figure 6g)
had a maximum at 19:00 h, which was slightly shifted to $R_{NO_3}^{\text{isoprene}}$ (Figure 6j). The evening peak
value of $R_{NO_3}^{\text{TVOCs}}$ could be accounted for by the accumulation of isoprene due to vegetation
emissions and traffic emissions that are released into the shallow nocturnal boundary layer. $R_{NO_3}^{\text{alkanes}}$
(Figure 7f), $R_{NO_3}^{\text{alkenes}}$ (Figure 7g), $R_{NO_3}^{\text{aromatics}}$ (Figure 7h), $R_{NO_3}^{\text{OVOCs}}$ (Figure 7i) and $R_{NO_3}^{SO_2}$
(Figure 6i) played virtually no roles in the diurnal variations of $R_{NO_3}^{\text{total}}$ and $R_{NO_3}^{\text{TVOCs}}$, although they
exhibited a more distinct diurnal profile.

**3.2.3 O₃ reactivity ($R_{O_3}$)**
The $R_{O_3}$ of trace gases was categorized into $NO_x$ and TVOCs which were grouped into alkanes,
alkenes, aromatics, OVOCs and isoprene (Table S1 lists the VOCs included in each group), as
shown in Figure 4e and 4f. The $R_{O_3}^{\text{total}}$ at the site varied between a minimum of $3.3\times10^{-4}\,\text{s}^{-1}$ and a
maximum of $1.8\times10^{-2}\,\text{s}^{-1}$ and was $1.2\pm1.7\times10^{-3}\,\text{s}^{-1}$ (± standard deviation) on average. NO exhibited
the most prominent contribution to the $R_{O_3}^{\text{total}}$ and represented on average >99% of the $R_{O_3}^{\text{total}}$,
whereas nearly all other contributions were < 1%. This result can be largely accounted for by the
generally large NO reaction rate coefficients with O₃ ($1.8\times10^{-14}\,\text{cm}^3\,\text{molecule}^{-1}\,\text{s}^{-1}$) (Atkinson et
al., 2006a), which are 3, >9, 2-4, >6, 4-6 and 3 orders of magnitude higher than the NO₂, alkanes,
alkenes, aromatics, OVOCs and isoprene reaction rate coefficients with NO₃, respectively (Atkinson
et al., 2006b;Atkinson and Arey, 2003;Yuan et al., 2013;Ferracci et al., 2018;Jenkin et al., 2015).
The majority of $R_{O_3}^{\text{total}}$ values were below $2\times10^{-3}\,\text{s}^{-1}$ but below $2\times10^{-6}\,\text{s}^{-1}$ and $2\times10^{-3}\,\text{s}^{-1}$ for
$R_{O_3}^{\text{TVOCs}}$ and $R_{O_3}^{NO_x}$, respectively, as seen in the frequency distribution (Figure S4j-S4l). The
cumulative frequency distribution clearly showed that the $R_{O_3}^{\text{total}}$ at low and high values was entirely
dominated by $R_{O_3}^{\text{TVOCs}}$ and $R_{O_3}^{NO_x}$, respectively (Figure S5c).



The $R_{O_3}$ of TVOCs, $R_{O_3}^{\text{TVOCs}}$, was $1.1\pm0.8\times10^{-6}$ s$^{-1}$ on average ranging from a minimum of
$2.5\times10^{-7}$ s$^{-1}$ to a maximum of $1.0\times10^{-5}$ s$^{-1}$. Alkenes clearly dominated the $R_{O_3}^{\text{TVOCs}}$ with campaign-
averaged contributions of 66%. Isoprene was the second largest contributor, comprising an average
of 28% of the $R_{O_3}^{\text{TVOCs}}$. In comparison, aromatics and OVOCs only accounted for 5% and 1%,
respectively, of the $R_{O_3}^{\text{TVOCs}}$. In contrast, the measured alkanes played nearly no role for $R_{O_3}^{\text{TVOCs}}$
due to their small reaction rate coefficients with O$_3$ ($<1.0\times10^{-23}$ cm$^3$ molecule$^{-1}$ s$^{-1}$) (Atkinson and
Arey, 2003;Atkinson et al., 2006b). The majority of $R_{O_3}^{\text{alkanes}}$, $R_{O_3}^{\text{alkenes}}$, $R_{O_3}^{\text{aromatics}}$, $R_{O_3}^{\text{OVOCs}}$ and
$R_{O_3}^{\text{isoprene}}$ were below $5.0\times10^{-12}$ s$^{-1}$, $2.0\times10^{-6}$ s$^{-1}$, $2.0\times10^{-7}$ s$^{-1}$, $2.0\times10^{-8}$ s$^{-1}$ and $2.0\times10^{-6}$ s$^{-1}$,
respectively (Figure S6k-S6o). The cumulative frequency distribution (Figure S9) clearly showed
that the $R_{O_3}^{\text{TVOCs}}$ at $>1.0\times10^{-7}$ was dominated by $R_{O_3}^{\text{alkenes}}$, $R_{O_3}^{\text{aromatics}}$ and $R_{O_3}^{\text{isoprene}}$,
the $R_{O_3}^{\text{TVOCs}}$ between $1.0\times10^{-9}$ and $1.0\times10^{-7}$ was dominated by $R_{O_3}^{\text{aromatics}}$, $R_{O_3}^{\text{isoprene}}$ and $R_{O_3}^{\text{OVOCs}}$,
and the $R_{O_3}^{\text{TVOCs}}$ $<1.0\times10^{-11}$ was entirely dominated by $R_{O_3}^{\text{alkanes}}$. In terms of individual species, cis-
2-butene, trans-2-butene, propylene, cis-2-pentene, styrene, ethylene and 1-butene were the top
seven species (Figure 3d), accounting for 25%, 20%, 7%, 6%, 5%, 5% and 3%, respectively, of
the $R_{O_3}^{\text{TVOCs}}$ and 0.4%, 0.2%, 1.2%, 0.1%, 0.5%, 5.3% and 0.5%, respectively, of the TVOC
emissions.

Compared with $R_{OH}$ and $R_{NO_3}$, $R_{O_3}$ displayed a much weaker diel variation, especially $R_{O_3}^{\text{alkenes}}$
and $R_{O_3}^{\text{aromatics}}$, as shown in Figure 6 and Figure 7. This can be explained by the following reasons.
First, for the same species, the reaction rate coefficients with O$_3$ were much smaller than its
corresponding reaction rate coefficients with OH and NO$_3$. For example, the ethylene reaction rate
coefficients with OH ($8.52\times10^{-12}$ cm$^3$ molecule$^{-1}$ s$^{-1}$) and NO$_3$ ($2.05\times10^{-16}$ cm$^3$ molecule$^{-1}$ s$^{-1}$) are
6 and 2 orders of magnitude higher, respectively, than the ethylene reaction rate coefficients with
O$_3$ ($1.59\times10^{-18}$ cm$^3$ molecule$^{-1}$ s$^{-1}$) (Atkinson and Arey, 2003;Atkinson et al., 2006b). Second, the
high-emissions species reaction rate coefficients with O$_3$ are smaller than the low-emissions species
reaction rate coefficients with O$_3$. For instance, the m/p-xylene (one of the top five emissions species)
reaction rate coefficients with O$_3$ ($<1.0\times10^{-20}$ cm$^3$ molecule$^{-1}$ s$^{-1}$) are much smaller than the 1-
hexene (one of the bottom five emissions species) reaction rate coefficients with O$_3$ ($1.13\times10^{-17}$ cm$^3$
molecule$^{-1}$ s$^{-1}$) (Atkinson and Arey, 2003;Atkinson et al., 2006b). The above two facets largely





weaken the diurnal variation in $R_{O_3}$.

### 3.3 $R_{OH}$ and O₃ production regimes

Photochemical formation is the main source of ground-level $O_3$, and VOCs, CO and $NO_x$ are key
precursors of tropospheric $O_3$ (Atkinson, 2000;Lyu et al., 2019). The production of $O_3$ is generally
limited by VOCs or $NO_x$ or is colimited by both VOCs and $NO_x$ (Lu et al., 2010;Tang et al., 2012;Li
et al., 2019a). However, $O_3$ formation is neither linearly dependent on $NO_x$ concentration nor VOC
reactivity (Pfannerstill et al., 2019); reductions in the emissions of these precursors can decrease,
increase, or leave the rate of $O_3$ production unchanged (Pusede and Cohen, 2012).

$R_{OH}^{VOCs}$ and $R_{OH}^{NO_x}$ are ways of defining $O_3$ production regimes (Kirchner et al., 2001;Lyu et al.,
2019;Pfannerstill et al., 2019;Sinha et al., 2012). In this study, we use the ratio of $R_{OH}^{VOCs}$ and $R_{OH}^{NO_x}$,
known as the s=$R_{OH}^{VOCs}/R_{OH}^{NO_x}$ ratio, to evaluate the $O_3$ production sensitivity, as suggested by
(Kirchner et al., 2001). The thresholds of the $R_{OH}^{VOCs}/R_{OH}^{NO_x}$ ratios are 0.2 and 0.01. When s > 0.2
indicates VOC limitation, 0.01<s<0.2 is colimited by both VOCs and $NO_x$, and s < 0.01 $NO_x$
limitation. The $O_3$ production regime plot (Figure 8) showed that Xianghe was characterized by a
strong VOC limitation. Here, 84% of the datapoints fall within the regime of VOC limitation,
whereas 16% are colimited by both VOCs and $NO_x$. The higher the $O_3$ concentration is, the more
obvious the VOC limitation will be. The lower the $O_3$ concentration is, the more obvious the
colimited by both VOCs and $NO_x$ will be. Previous studies based on space-based $HCHO/NO_2$
column ratio (Tang et al., 2012) and VOC / $NO_x$ ratio (Wang et al., 2014b) also found that summer
$O_3$ production in this district may be under a VOC-limited regime. In addition, as VOCs generally
have good correlations with CO and play a similar role as CO in photochemical $O_3$ production
(Atkinson, 2000). A scatter plot of CO-$NO_y$ can also be used to evaluate the $O_3$-$NO_x$-VOC
sensitivity (Ding et al., 2013). Figure S10 depicts the scatter plots of CO-$NO_y$ color-coded with $O_3$
concentrations. The results showed that high $O_3$ levels are generally associated with a high CO/$NO_y$
ratio, indicating a VOC-limited regime of $O_3$ formation in Xianghe. Generally, our results suggested
that control of VOCs would be most effective for controlling $O_3$ in Xianghe.



### 3.4 Implications for $R_{OH}$, $R_{NO_3}$ and $R_{O_3}$-based VOC control strategies

Table 1 lists the top 10 VOC species (excluded isoprene) in terms of concentrations, $R_{OH}$, $R_{NO_3}$ and $R_{O_3}$, and their corresponding contributions to concentrations, $R_{OH}$, $R_{NO_3}$ and $R_{O_3}$. The order of the major $R_{OH}$, $R_{NO_3}$ and $R_{O_3}$-contributing species differed significantly from that of concentration-contributing species. Therefore, VOC control strategies based on $R_{OH}$, $R_{NO_3}$ and $R_{O_3}$ differ significantly from those based on concentrations.

From the perspective of concentrations, propane, acetone, ethane, n-butane, m/p-xylene, iso-pentane, ethylene, iso-butane, n-pentane and toluene should be targeted. If these 10 species were fully controlled, it would lead to a VOC concentration reduction of 74.1% with only $R_{OH}$, $R_{NO_3}$ and $R_{O_3}$ reductions of 43.1%, 0.4% and 6.4%, respectively. From the perspective of $R_{OH}$, m/p-xylene, ethylene, hexanal, o-xylene, propylene, styrene, methacrolein, cis-butene, methylvinyketone and iso-pentane were the key species. If releases of these compounds were reduced to zero without any offset, it would reduce $R_{OH}$ by 65.1% with a VOC concentration reduction of 24.9%, a $R_{NO_3}$ reduction of 87.1% and a $R_{O_3}$ reduction of 57.1%. From the perspective of $R_{NO_3}$, the top 10 VOC species consisted of styrene, cis-2-butene, trans-2-butene, cis-2-pentene, hexanal, propylene, 1,3-butadiene, 1-butene, trans-2-pentene and pentanal. If the concentrations of these species were completely eliminated, it would reduce $R_{NO_3}$ by 98.0% with a VOC concentration reduction of 4.5%, a $R_{OH}$ reduction of 28.7% and a $R_{O_3}$ reduction of 91.4%. From the perspective of $R_{O_3}$, cis-2-butene, trans-2-butene, propylene, cis-2-pentene, styrene, ethylene, 1-butene, trans-2-pentene, 1-pentene and methacrolein should be key targets for control. If the concentrations of these compounds were reduced to zero without any offset, it would lead to a $R_{O_3}$ reduction of 98.9% with a VOC concentration reduction of 9.1%, a $R_{OH}$ reduction of 31.9% and a $R_{NO_3}$ reduction of 95.6%. Clearly, species with large concentrations do not necessarily have high $R_{OH}$, $R_{NO_3}$ and $R_{O_3}$, and with the least concentration reduction, the maximum reduction of activity can be obtained. The key VOC species of $R_{OH}$, $R_{NO_3}$ and $R_{O_3}$ also differed from each other. However, reducing concentrations of propylene, styrene and cis-butene may likely achieve a win-win-win situation. Although the above comparisons were made under the assumption that concentrations would be significantly reduced, it is obvious that $R_{OH}$, $R_{NO_3}$ and $R_{O_3}$-based control strategies are more efficient than concentration-based policies in terms of reducing VOC pollution.





615

**3.5 Atmospheric oxidation capacity (AOC)**

**3.5.1 Overall characteristics of AOC**

The loss rate of VOCs and CO via reactions with OH, $O_3$ and $NO_3$ was calculated. The calculated AOC was up to $4.4 \times 10^8$ molecules cm$^{-3}$ s$^{-1}$ with campaign-averaged values of $3.1 \times 10^7$ molecules cm$^{-3}$ s$^{-1}$, daytime averages (06:00-18:00 LT) of $5.2 \times 10^7$ molecules cm$^{-3}$ s$^{-1}$ and nighttime averages of $1.5 \times 10^6$ molecules cm$^{-3}$ s$^{-1}$. As such, the total number of CO and VOC molecules depleted during daytime and nighttime were $2.4 \times 10^{12}$ and $6.1 \times 10^{10}$, respectively, per cm$^{-3}$ of air. Such AOC levels were lower than those determined at the Tung Chung air quality monitoring station (Xue et al., 2016) and from a polluted area in Santiago, Chile (Elshorbany et al., 2009), but comparable to that determined at the Hong Kong Polytechnic University's air monitoring station at Hok Tsui (Li et al., 2018).

Comparisons of calculated AOC by OH, $O_3$ and $NO_3$ and corresponding oxidation concentrations are shown in Figure 9. The calculated AOC by OH, $O_3$ and $NO_3$ followed well with the corresponding oxidation concentrations, with correlation coefficients (r) of 0.96, 0.55 and 0.88, respectively, suggesting that the calculated AOC here was consistent with the one obtained using radical concentration to indicate AOC. Specifically, the average oxidation capacities of OH (Figure 9a), $O_3$ (Figure 9b) and $NO_3$ (Figure 9c) radicals throughout the entire campaign were $2.9 \times 10^7$, $1.2 \times 10^6$ and $1.7 \times 10^5$ molecule cm$^{-3}$ s$^{-1}$, representing 95.4, 4.0 and 0.6% of the total oxidation capacity, respectively. The total number of depleted molecules per day due to oxidation by OH, $O_3$ and $NO_3$ were $2.5 \times 10^{12}$, $1.1 \times 10^{11}$ and $1.5 \times 10^{10}$ molecules cm$^{-3}$, respectively, which was slightly lower than that assessed from a polluted area in Santiago, Chile (Elshorbany et al., 2009). Accordingly, the OH radical is the driving force of the oxidation capacity in the atmosphere in Xianghe, especially during daytime. Figure 10 shows a comparison of the oxidation capacities of OH, $O_3$ and $NO_3$. OH is the only oxidant for CO in the troposphere. As expected, OH was responsible for 97% of the oxidation capacity regarding VOCs and CO during the daytime (Figure 10a). The relative contribution of OH initiating oxidation capacity decreased to 94% when restricting the calculation to VOC families alone (Figure 10b). Focusing on the oxidation of unsaturated VOC, OH was the dominant oxidant with a relative proportion of approximately 100%


(Figure 10c). Note that the influence of $NO_3$ and $O_3$ on the oxidation of CO and VOCs can be
neglected during the daytime. However, the elevated relative contributions of $O_3$ and $NO_3$ initiating
oxidation capacity can be observed during nighttime. As expected, $O_3$ and $NO_3$ accounted for 58%
and 11%, respectively, of the oxidation capacity regarding VOCs and CO (Figure 10d), but 67%
and 13% of VOC families alone (Figure 10e) occurred at night. Focusing on the oxidation of
unsaturated VOC, $O_3$ and $NO_3$ accounted for 68% and 13%, respectively, of the oxidation capacity
(Figure 10f). Compared with OH and $O_3$, $NO_3$ had a lower contribution during both the daytime and
nighttime, which was mainly caused by the high NO concentrations (Liebmann et al., 2018b).

**3.5.2 Relative contributions of VOC oxidation pathways**
VOCs are mainly consumed by reactions with OH radicals, $O_3$ and $NO_3$ radicals in the atmosphere
(Tang et al., 2017;Atkinson and Arey, 2003;Yuan et al., 2013;Vereecken and Francisco, 2012). The
time series of VOC loss rates due to the reactions with OH radicals, $O_3$ and $NO_3$ radicals are depicted
in Figure 11. Diurnal variations of VOC groups and individual species loss rates due to the reactions
with different oxidants are shown in Figure 12 and Figure S11-S14, respectively. A comparison of
the relative contribution of OH, $NO_3$ and $O_3$ to the daytime and nighttime integral of the oxidation
rate is illustrated in Figure 13.

Reactions with OH radicals were the dominant losses for alkanes, accounting for approximately
100% and 99% of the daytime and nighttime average loss rates of alkanes, respectively. In contrast,
reactions with $O_3$ and $NO_3$ were nonsignificant for the loss rates of alkanes, accounting for <1% of
both the daytime and nighttime average loss rates of alkanes.

Since alkenes have a greater reaction rate with $O_3$, oxidation by $O_3$ also contributes to the loss rates
of alkenes. Oxidation by $O_3$ accounted for 24% of the daytime average total loss rate of alkenes and
increased to 94% during nighttime. Specifically, the reaction with $O_3$ was the dominant contributor
to loss rates of trans-2-butene, cis-2-butene and trans-2-pentene, with daytime contributions of 63,
51 and 56% and nighttime contributions of 91, 87 and 89%, respectively. The relative contributions
of $O_3$ to the nighttime integral of the oxidation rates of propylene, 1-butene, 1-pentene and 1-hexene
were 61, 54, 72 and 62%, respectively. Reaction oxidation by OH radicals dominated the daytime





and nighttime integral of the loss rates of the rest of the species including ethylene, 1,3-butadiene,
cis-2-pentene and isoprene. Significantly, in contrast to anthropogenic hydrocarbons, the oxidation
by the $NO_3$ radical is more important for the loss rates of isoprene, contributing to <1% and 22% of
the daytime and nighttime average loss rates of isoprene, respectively. The contribution of the 24 h
average loss rates of isoprene oxidized by $NO_3$ (14%) was much lower than that determined in the
Changdao campaign (26%) (Yuan et al., 2013), which was probably caused by the higher $O_3$
concentrations in this study.

For most OVOC species, the reactions with OH radicals were the only significant contributor to
OVOC loss rates except for acetone, where the reaction with $O_3$ accounted for 57% of the nighttime
average loss rates of acetone. Similar to OVOC species, the reactions with OH radicals were also
the only significant contributor to aromatic loss rates, except for styrene, where the reaction with
$O_3$ and $NO_3$ accounted for 47% and 46%, respectively, of the nighttime average loss rates of styrene.
In total, oxidation by OH radicals accounted for approximately 100% and 81% of the daytime and
nighttime average loss rates of OVOCs, respectively. Oxidation by OH radicals, $NO_3$ radicals and
$O_3$ accounted for 97, 2 and 1%, respectively, of daytime average loss rates of aromatics, whereas
during the nighttime, the contributions from the reactions with OH radicals, $NO_3$ radicals and $O_3$
were 33, 33 and 34%, respectively.

We also emphasized that the concentrations of $NO_3$ are not only influenced by VOCs but also by
NO and the heterogeneous loss of $N_2O_5$ (Liebmann et al., 2018b;Yuan et al., 2013;Crowley et al.,
2011;Sobanski et al., 2016;Geyer et al., 2001). In this study, $NO_3$ loss due to $N_2O_5$ hydrolysis was
not accounted for in Eq. (6). The predicted stationary-state $NO_3$ concentrations calculated from Eq.
(6) were upper limits, and hence, the calculation of $NO_3$ contributions to VOC losses was also
overestimated.

**4 Conclusions**
In the summer of 2018, a comprehensive field campaign was conducted at a suburban site in the
North China Plain. Based on simultaneous measurements of $O_3$, CO, $SO_2$, NO, $NO_2$, $J_{O^1D}$, $J_{NO_2}$,
$J_{NO_3}$ and 65 VOCs, reactivities (OH, $NO_3$ and $O_3$ reactivities) for trace gases and atmospheric



oxidation capacity (AOC) were comprehensively analyzed. The main findings are summarized as
follows.

The total, $R_{OH}^{\text{total}}$, was between 8.5 and 68.1 s$^{-1}$ with an average of 25.6±9.7 s$^{-1}$, which was mainly
contributed by NO$_x$ (12.0±7.1 s$^{-1}$, 47%), followed by CO (7.2±2.6 s$^{-1}$, 28%) and TVOCs (6.2±4.6
s$^{-1}$, 24%) and to a lesser extent by SO$_2$ and O$_3$ (0.2±0.1 s$^{-1}$, 1%). $R_{OH}^{\text{TVOCs}}$ was 6.2±4.6 s$^{-1}$ and
dominated by isoprene. Campaign-averaged values of $R_{NO_3}^{\text{total}}$ were 2.2±2.6 s$^{-1}$, ranging from 0.7
s$^{-1}$ to 27.5 s$^{-1}$. NO$_x$ was by far the main contributors to the $R_{NO_3}^{\text{total}}$, representing 99% of the $R_{NO_3}^{\text{total}}$
on average. $R_{NO_3}^{\text{TVOCs}}$ was 2.2±2.7×10$^{-2}$ s$^{-1}$ on average with a minimum of 1.0×10$^{-3}$ s$^{-1}$ and a
maximum of 0.2 s$^{-1}$. The largest fraction of attributed $R_{NO_3}^{\text{TVOCs}}$ was contributed by isoprene (77%).
The $R_{O_3}^{\text{total}}$ varied between a minimum of 3.3×10$^{-4}$ s$^{-1}$ and a maximum of 1.8×10$^{-2}$ s$^{-1}$ and was
1.2±1.7×10$^{-3}$ s$^{-1}$ on average. NO exhibited the most prominent contribution to the $R_{O_3}^{\text{total}}$ and
represented an average of >99% of the $R_{O_3}^{\text{total}}$. $R_{O_3}^{\text{TVOCs}}$ was 1.1±0.8×10$^{-6}$ s$^{-1}$ on average, ranging
from 2.5×10$^{-7}$ s$^{-1}$ to 1.0×10$^{-5}$ s$^{-1}$. Alkenes clearly dominated the $R_{O_3}^{\text{TVOCs}}$ with campaign-averaged
contributions of 66%.

$R_{OH}^{\text{total}}$, $R_{NO_3}^{\text{total}}$ and $R_{O_3}^{\text{total}}$ displayed a similar diel variation with the lowest in the afternoon and the
highest during rush hours, and the diurnal profile of NO$_x$ appears to be the major driver for the
diurnal profiles of $R_{OH}^{\text{total}}$, $R_{NO_3}^{\text{total}}$ and $R_{O_3}^{\text{total}}$. Compared with $R_{OH}$ and $R_{NO_3}$, $R_{O_3}$ displayed a
much weaker diel variation, especially $R_{O_3}^{\text{alkenes}}$ and $R_{O_3}^{\text{aromatics}}$ due to 1) the rate coefficients with
O$_3$ being much smaller than the corresponding reaction rate coefficients with OH and NO$_3$ for the
same species and 2) the high-emissions species reaction rate coefficients with O$_3$ being smaller than
the low-emissions species reaction rate coefficients with O$_3$.

The $R_{OH}^{\text{VOCs}}/R_{OH}^{\text{NO}_x}$ ratio and scatter plots of CO-NO$_y$ color-coded with O$_3$ concentrations indicated
a VOC-limited regime of O$_3$ formation in Xianghe, suggesting that control of VOCs would be most
effective for controlling O$_3$ in Xianghe. $R_{OH}$, $R_{NO_3}$ and $R_{O_3}$-based control strategies are more
efficient than concentration-based policies in terms of reducing VOC pollution. We suggest that
policy makers shift the current concentration -based limits to reactivity-based policies.



The loss rates of VOCs and CO via reactions with OH, $O_3$ and $NO_3$ were calculated, which were up
to $4.4 \times 10^8$ molecules $cm^{-3}$ $s^{-1}$ with campaign-averaged values of $3.1 \times 10^7$ molecules $cm^{-3}$ $s^{-1}$,
daytime averages (06:00-18:00 LT) of $5.2 \times 10^7$ molecules $cm^{-3}$ $s^{-1}$ and nighttime averages of
$1.5 \times 10^6$ molecules $cm^{-3}$ $s^{-1}$. The AOC was dominated by OH radicals ($2.9 \times 10^7$ molecule $cm^{-3}$ $s^{-1}$,
95%), $O_3$ ($1.2 \times 10^6$ molecule $cm^{-3}$ $s^{-1}$, 4%) and $NO_3$ radicals ($1.7 \times 10^5$ molecule $cm^{-3}$ $s^{-1}$, 1%),
suggesting that the OH radical is the driving force of the oxidation capacity in the atmosphere in
Xianghe, especially during the daytime. The reaction with OH radicals was the dominant loss for
VOCs except for trans-2-butene, cis-2-butene, trans-2-pentene, propylene, 1-butene, 1-pentene, 1-
hexene, acetone and styrene, where the reaction with $O_3$ was more important for their loss rates.
Compared with anthropogenic hydrocarbons, the oxidation by $NO_3$ radical was more important for
the nighttime integral of isoprene loss rates.

Our study provides useful insights for VOC pollution control in a typical suburban site in the North
China Plain. Further studies, especially direct observations of the OH and $NO_3$ radical, OH and $NO_3$
reactivity measurements and speciated measurements, are required to further explore the trace gas
reactivity and AOC.

**Acknowledgement**

753         This study was financially supported by the Ministry of Science and Technology of China

(2017YFC0210000), Beijing Major Science and Technology Project (Z181100005418014). All
referenced supplemental figures and tables can be found in the supporting information. The authors
are grateful to all staff and workers from the Xianghe Atmospheric Observatory of Institute of
Atmospheric Physics (IAP) of the Chinese Academy of Sciences for their support during the
sampling campaign. We also acknowledge National Meteorological Information Center for
providing high quality meteorology parameters.

**Competing financial interests**
The authors declare no competing financial interests.



**Author contributions**

Y.S.W designed the research. Y.Y and D.Y, S.M.Z, D.S.J, Y.H.W conducted the measurements. Y.Y and Y.H.W interpreted the data and write the paper. All the authors commented on the paper.

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




**Figure captions**



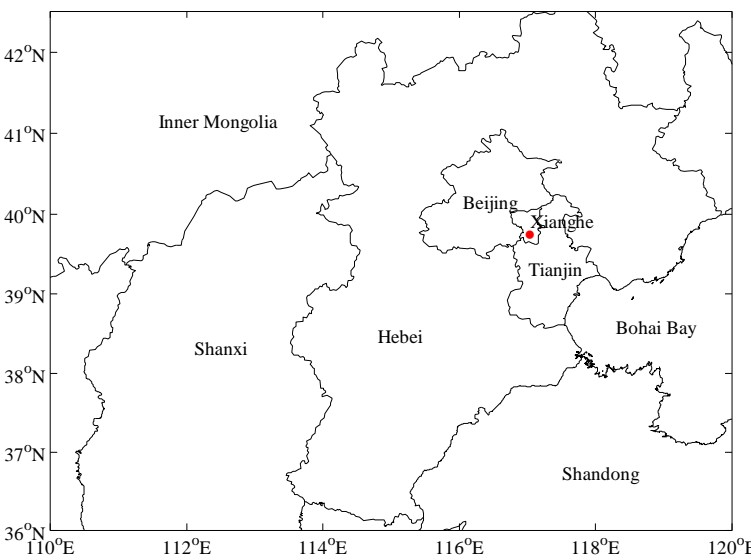

Figure 1. The location of the sampling site, which is marked with a red dot. The blacklines are
provincial boundary lines of each province. (The figure was produced by MATLAB 2017a).















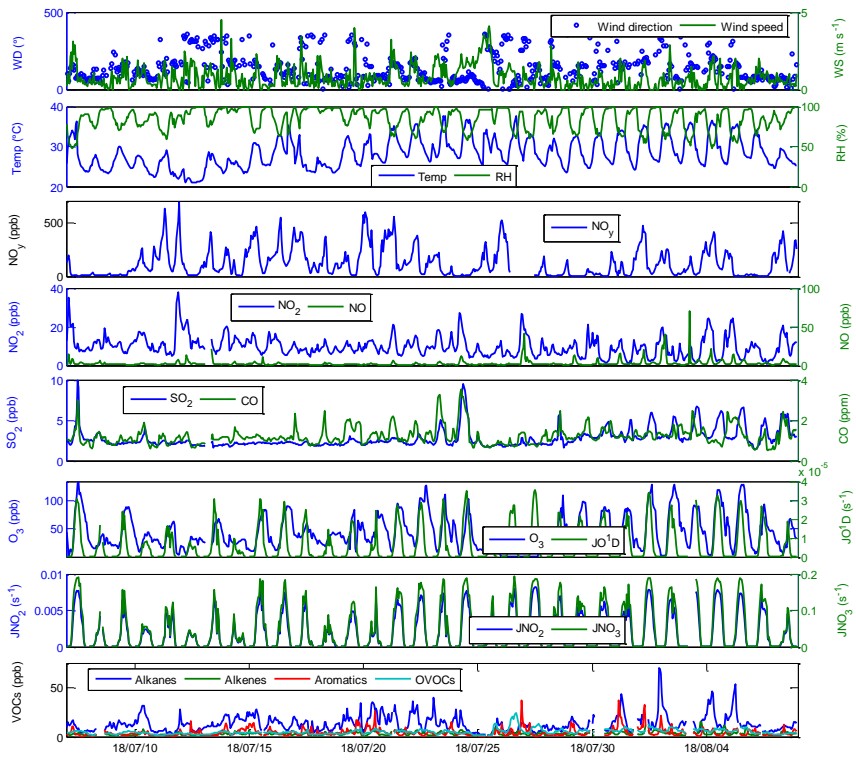


Figure 2. Time series of meteorology parameters, trace gases, photolysis rates of JNO$_2$ and JNO$_3$,
and VOCs concentrations during the field campaign at Xianghe from 6 July to 6 August 2018.









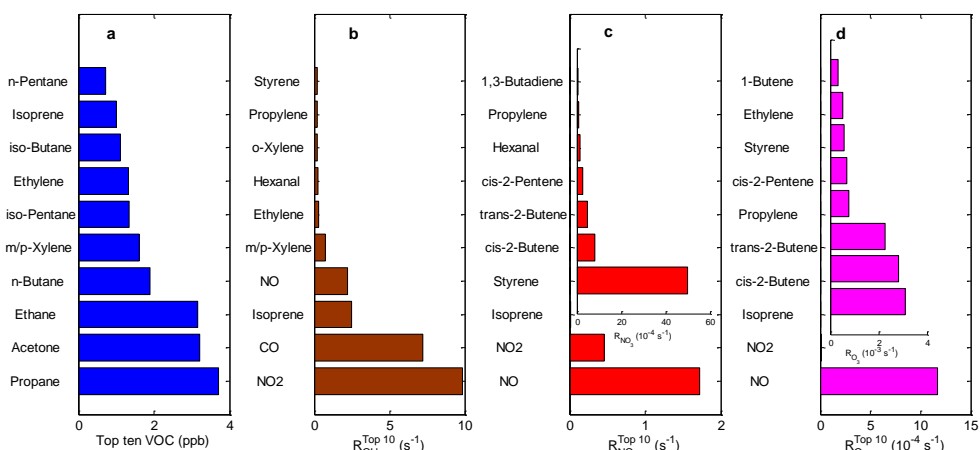


Figure 3. The top ten abundant VOC species (a), reactivity of OH ($R_{OH}^{\text{calculated}}$) (b), reactivity of NO$_3$
($R_{NO_3}^{\text{calculated}}$)  (c) and reactivity of O$_3$  ($R_{O_3}^{\text{calculated}}$)  (d) during the field campaign at Xianghe from
6 July to 6 August 2018.











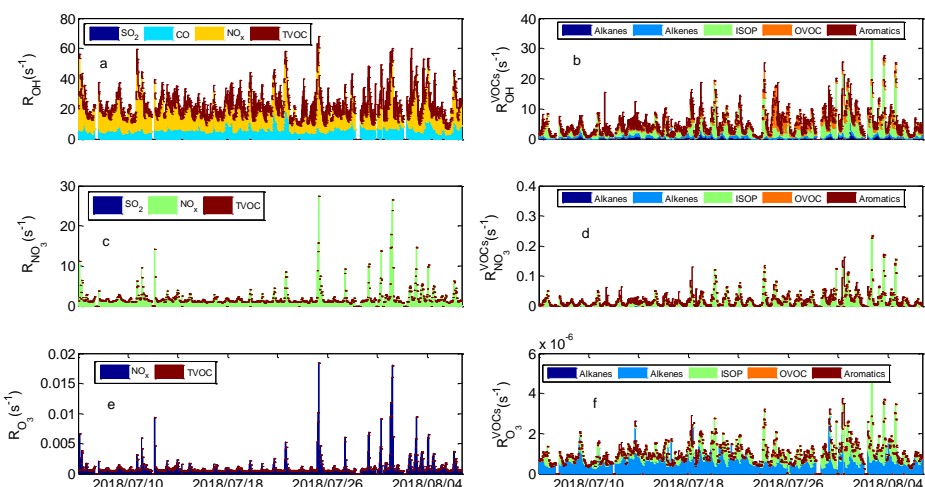


Figure 4. The time series of $R_{OH}^{calculated}$, $R_{NO_3}^{calculated}$ and $R_{O_3}^{calculated}$ during the field campaign at

Xianghe from 6 July to 6 August 2018.











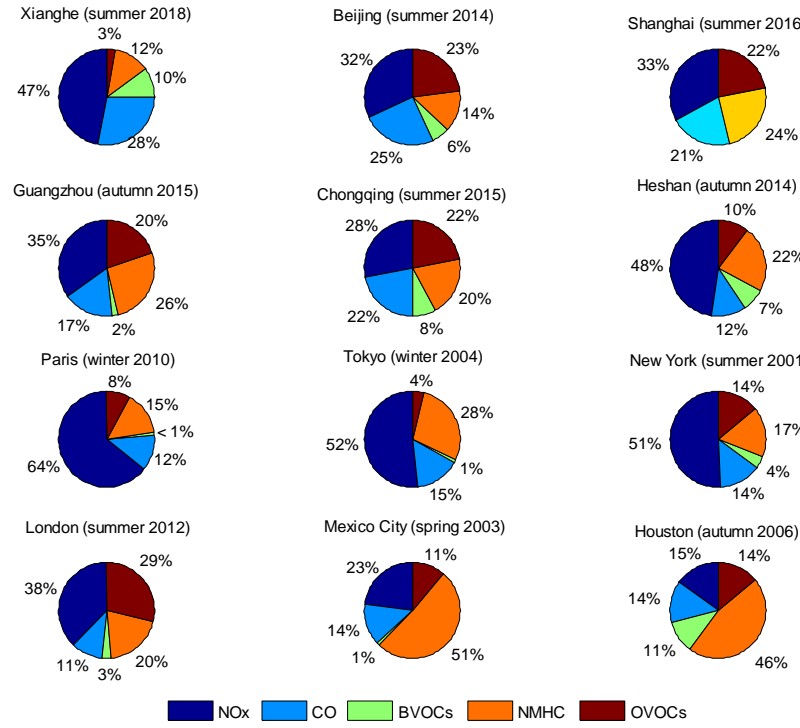

Figure 5. Contributions of different atmospheric compounds to $R_{OH}$ at Xianghe (summer 2018,

this study), Beijing (summer 2014, Tan et al., 2019), Shanghai (summer 2016, Tan et al., 2019),

Guangzhou (autumn 2015, Tan et al., 2019), Chongqing (summer 2015, Tan et al., 2019), Heshan

(autumn 2014, Yang et al., 2017), Paris (winter 2010, Dolgorouky et al., 2012), Tokyo (winter 2004,

Yoshino et al., 2006), New York (summer 2001, Mao et al., 2010), London (summer 2012, Whalley

et al., 2016), Mexico City (spring 2003, Mao et al., 2010) and Houston (autumn 2006, Mao et al.,

2010).











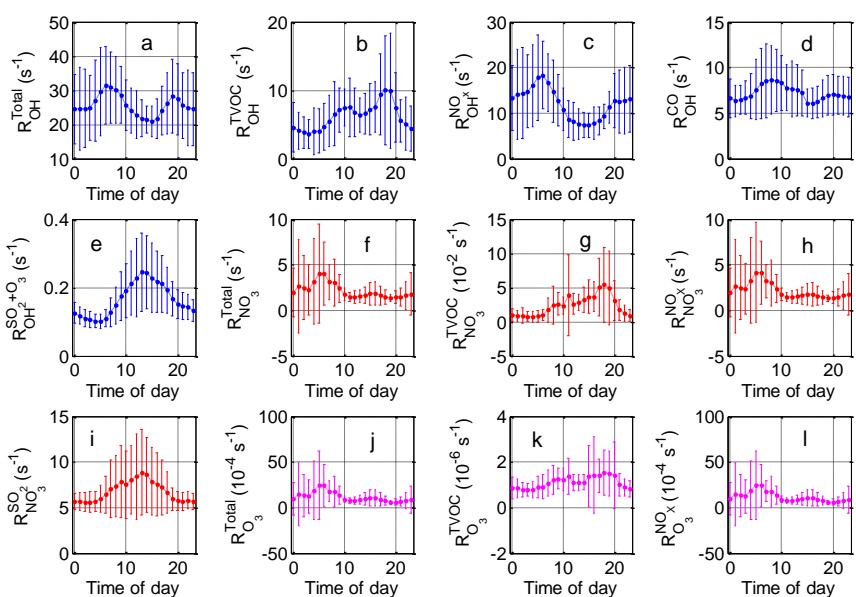

Figure 6. Mean diurnal variations of $R_{OH}^{calculated}$, $R_{NO_3}^{calculated}$ and $R_{O_3}^{calculated}$ of trace gases during
the field campaign at Xianghe from 6 July to 6 August 2018.















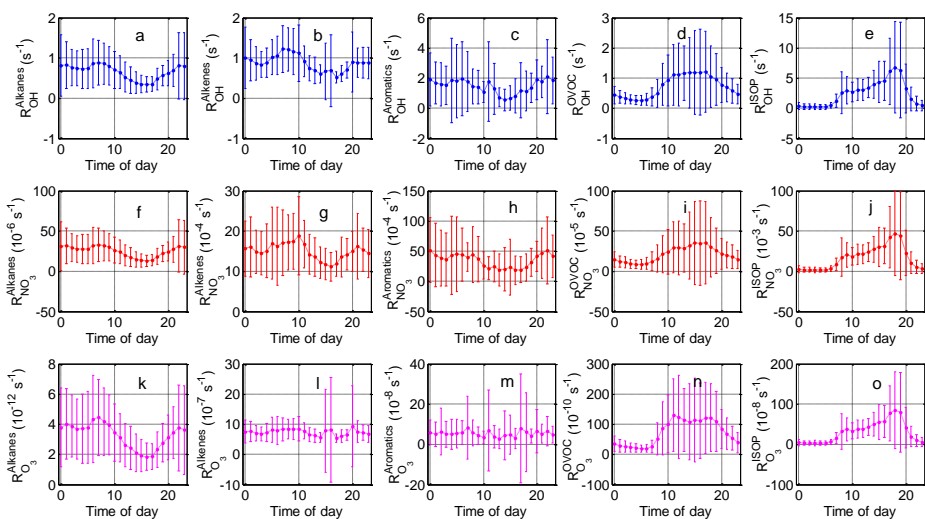

Figure 7. Mean diurnal variations of $R_{OH}^{calculated}$, $R_{NO_3}^{calculated}$ and $R_{O_3}^{calculated}$ of VOC groups
during the field campaign at Xianghe from 6 July to 6 August 2018.















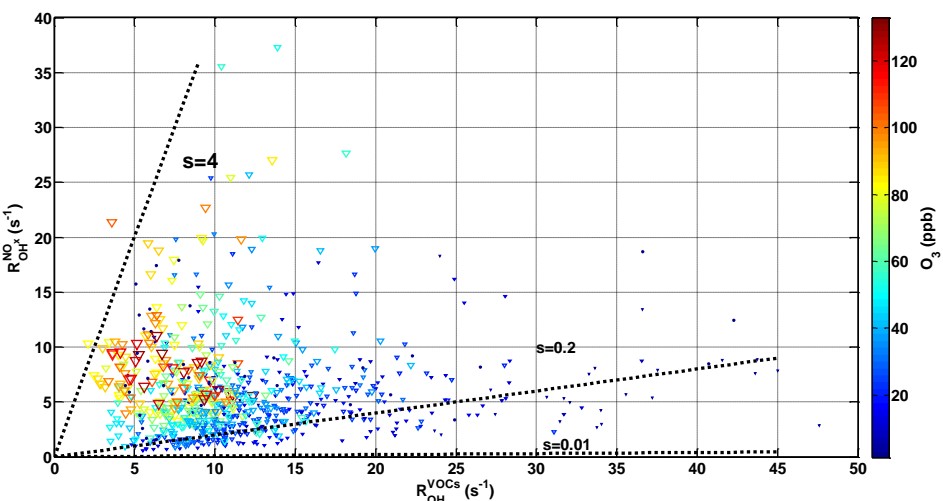

Figure 8. O$_3$ production regimes at Xianghe station. The dot lines are the borders of the three regimes
of O$_3$ formation. "s" denotes the relative reactivity of OH towards NO$_x$ and VOCs. For s > 0.2: VOC
limitation, for s < 0.01: NO$_x$ limitation of the O$_3$ formation.














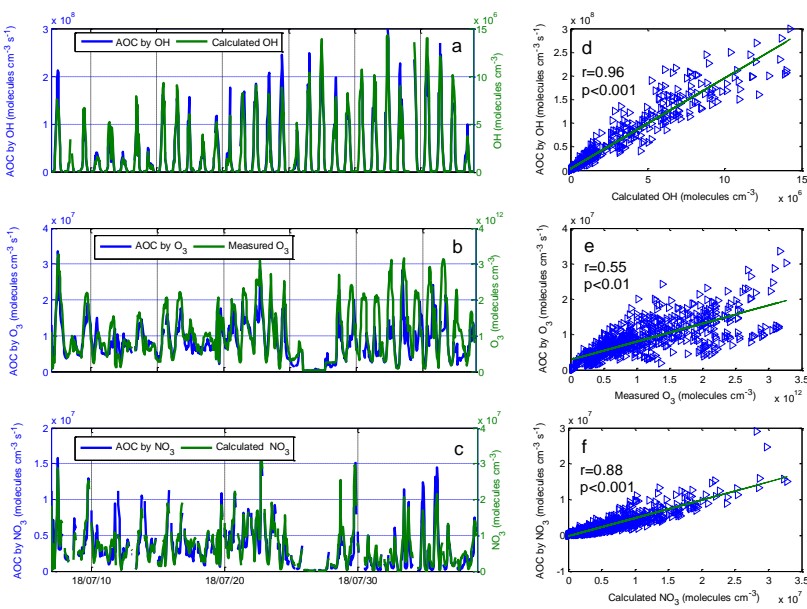

Figure 9. Comparisons of calculated AOC by OH (a), $O_3$ (b) and $NO_3$ (c), and corresponding
oxidation concentrations. The left column shows the time series and the right column shows
scatterplots of calculated AOC and corresponding oxidation concentrations. Note: r and p are the
correlation coefficient and the significance level, respectively.













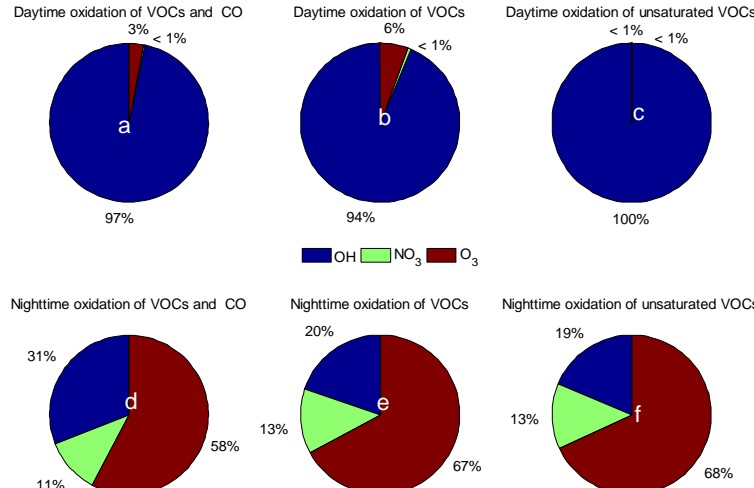

Figure 10. Comparison of the relative contributions of OH, NO$_3$ and O$_3$ to the daytime and nighttime
integral of the oxidation rates. Data are calculated for the oxidation of (a,d) VOCs and CO, (b,e)
VOCs only, and (c,f) unsaturated VOCs only.











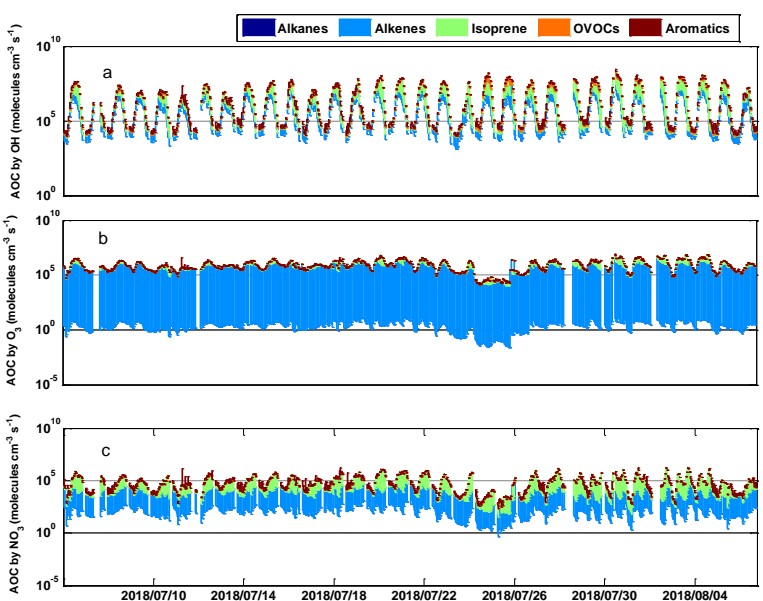

Figure 11.The time series of VOC loss rates due to the reactions with OH radical, $O_3$ and $NO_3$
radical.















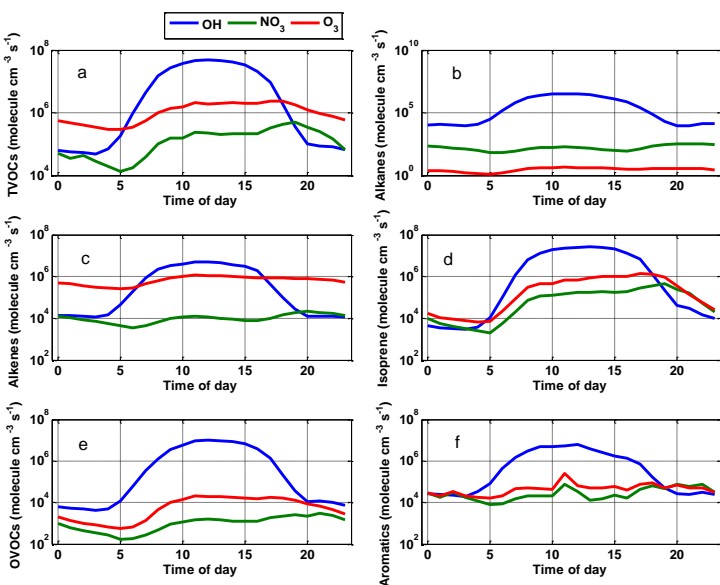

Figure 12. Diurnal variations of VOC loss rates due to the reactions with OH radical (blue lines),
NO$_3$ radical (green lines) and O$_3$ (red lines).













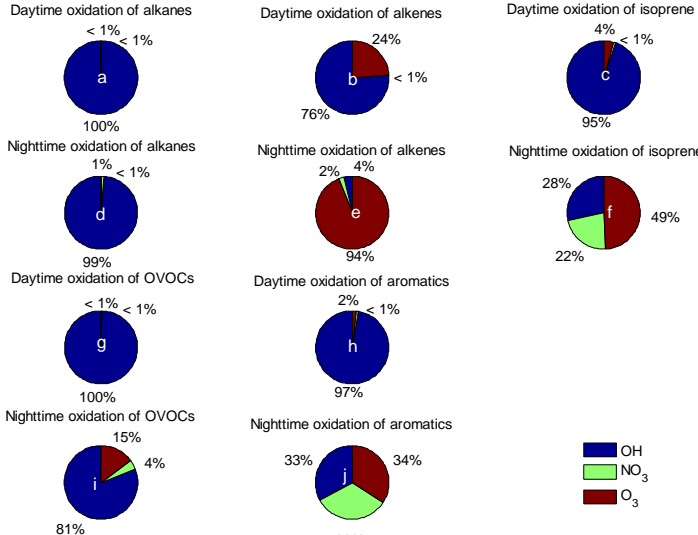

Figure 13. Comparison of the relative contributions of OH, NO₃ and O₃ to the daytime and nighttime
integral of the oxidation rates. Data are calculated for the oxidation of (a,d) alkanes, (b,e) alkenes,
(c,f) isoprene, (g,i) OVOCs and (h,j) aromatics.














**Table captions**
Table 1. The top 10 VOCs species in terms of concentration (first column), $R_{OH}$ (second column),
$R_{NO_3}$ (third column) and $R_{O_3}$ (fourth column) and their corresponding contributions to
concentration, $R_{OH}$, $R_{NO_3}$ and $R_{O_3}$ (%).

| First column | | | | | Second column | | | | |
|---|---|---|---|---|---|---|---|---|---|
| Species | Concentration | $R_{OH}$ | $R_{NO_3}$ | $R_{O_3}$ | Species | Concentration | $R_{OH}$ | $R_{NO_3}$ | $R_{O_3}$ |
| Propane | **14.6** | 2.7 | 0.1 | 0.0 | m/p-Xylene | 6.4 | **20.1** | 0.1 | 0.0 |
| Acetone | **12.7** | 0.4 | 0.0 | 0.1 | Ethylene | 5.3 | **7.3** | 0.1 | 6.3 |
| Ethane | **12.5** | 0.5 | 0.0 | 0.0 | Hexanal | 1.3 | **6.6** | 1.9 | 0.0 |
| n-Butane | **7.5** | 3.0 | 0.0 | 0.0 | o-Xylene | 2.6 | **5.8** | 0.1 | 0.0 |
| m/p-Xylene | **6.4** | 20.1 | 0.1 | 0.0 | Propylene | 1.2 | **5.4** | 1.0 | 9.3 |
| iso-Pentane | **5.3** | 3.2 | 0.1 | 0.0 | Styrene | 0.5 | **5.2** | 72.0 | 6.8 |
| Ethylene | **5.3** | 7.3 | 0.1 | 6.3 | Methacrolein | 0.8 | **3.9** | 0.2 | 0.7 |
| iso-Butane | **4.4** | 1.6 | 0.0 | 0.0 | cis-2-Butene | 0.4 | **3.9** | 11.5 | 34.0 |
| n-Pentane | **2.8** | 1.8 | 0.0 | 0.0 | MethylVinylKetone | 1.1 | **3.7** | 0.1 | 0.0 |
| Toluene | **2.6** | 2.5 | 0.0 | 0.0 | iso-Pentane | 5.3 | **3.2** | 0.1 | 0.0 |
| Third column | | | | | Forth column | | | | |
| Species | Concentration | $R_{OH}$ | $R_{NO_3}$ | $R_{O_3}$ | Species | Concentration | $R_{OH}$ | $R_{NO_3}$ | $R_{O_3}$ |
| Styrene | 0.5 | 5.2 | **72.0** | 6.8 | cis-2-Butene | 0.4 | 3.9 | 11.5 | **34.0** |
| cis-2-Butene | 0.4 | 3.9 | **11.5** | 34.0 | trans-2-Butene | 0.2 | 1.8 | 6.7 | **27.3** |
| trans-2-Butene | 0.2 | 1.8 | **6.7** | 27.3 | Propylene | 1.2 | 5.4 | 1.0 | **9.3** |
| cis-2-Pentene | 0.1 | 0.9 | **2.8** | 8.1 | cis-2-Pentene | 0.1 | 0.9 | 2.8 | **8.1** |
| Hexanal | 1.3 | 6.6 | **1.9** | 0.0 | Styrene | 0.5 | 5.2 | 72.0 | **6.8** |
| Propylene | 1.2 | 5.4 | **1.0** | 9.3 | Ethylene | 5.3 | 7.3 | 0.1 | **6.3** |
| 1,3-Butadiene | 0.1 | 0.9 | **0.7** | 0.4 | 1-Butene | 0.5 | 2.7 | 0.6 | **3.7** |
| 1-Butene | 0.5 | 2.7 | **0.6** | 3.7 | trans-2-Pentene | 0.0 | 0.2 | 0.5 | **1.8** |
| trans-2-Pentene | 0.0 | 0.2 | **0.5** | 1.8 | 1-Pentene | 0.1 | 0.6 | 0.2 | **0.9** |
| Pentanal | 0.2 | 1.1 | **0.3** | 0.0 | Methacrolein | 0.8 | 3.9 | 0.2 | **0.7** |
