# Peer review of "Parameterized atmospheric reactivity and oxidation capacity during summer at a"

_Atmospheric Chemistry and Physics, 2019_

## Referee Comment (RC1) · Anonymous Referee #2 · 30 Dec 2019

Oxidation capacity is an important parameter to understand the atmospheric chemistry of air pollutants. This work analyzed the ROH, RO3 and RNO3 based on the measured VOCs and traditional trace gases concentrations in Xiang He. Overall, the methods are reasonable and the data are robust. After the following questions have been well addressed, it is publishable.

1. Isoprene is also an alkene. I understand the authors want to differentiate the anthropogenic VOCs from the biogenic VOCs. I suggest to define them more strictly and accurately. 2. In equations 1-3, the "k" should be lower case letters for rate constant.

[Figure]

3. When calculating the reactivity, did you consider the influence of temperature on the rate constants? How about the uncertainties for the calculation? Can you give a comment on the possible difference for the measured R and estimated R? 4. Although the authors compared the calculated R values with different places. It is difficult to follow it in the text. I suggest to list them in a supplement table. 5. Traffic is not the only source of NOx. Thus, it is not reasonable to ascribe the ROH to traffic Line 385. 6. When comparing the ROH(TVOCs) with other researches, the comparison of VOCs composition is necessary among different researchers (lines 399-419). 7. When discussing the implication for control strategies, I think it is more reasonable to normalize the reactivity to secondary pollutants formation potential.

---

## Referee Comment (RC2) · Anonymous Referee #3 · 6 Jan 2020

This paper shows OH, NO3, and O3 reactivity from VOC and traces gas measurements conducted in Xianghe in 2018 from 6 July to 6 August. In addition, the authors estimate the trace gases oxidation rate using parametrized OH, NO3, and observed O3 concentrations, which is defined as oxidation atmospheric oxidation capacity. This data set helps to add to the increasing knowledge of the oxidant reactivity. The atmospheric oxidation capacity highly depends on the parametrization. Though this method is not new, a detail uncertainty analysis related to the calculation is missing. This reviewer suggests using a box model to calculate the OH and NO3 concentrations or prove the

justification of the parameterization. Besides, it's difficult to follow the writing, especially the authors tried to compare their results with other campaigns. The manuscript needs a significant reduction to be concise and informative before reconsidering.

Specific comments: 1. Line 266-270, It's not clear which values are used from which literature. If there is difference between different literatures, e.g. OH+NO2, which one is used? 2. Line 270. Why not use the newest version of Master Chemical Mechanism v3.3.1 3. OH is parameterized to jo1d, jno2, and NO2 using the results from a rural site in Germany, which could be different from the present study. A box/regional model to simulate OH concentration is helpful to validate the parameterization at Xianghe. On the other hand, previous field OH observations in China demonstrate that a strong correlation exists between OH and jO1D with a relatively constant slope $4.5\pm0.5\times10^{11}$cm-3 s-1 (Lu et al. 2012 10.5194/acp-13-1057-2013; Tan et al. (2017) 10.5194/acp-17-663-2017; Tan et al. (2018) 10.5194/acp-18-12391-2018). Maybe it's also a good idea to show the parametrized OH concentrations in supplement. 4. The parameterization of NO3 is improved by considering the conversion to N2O5 compared to the first version. A proper discussion related to this uncertainty is missing. In equation (4), AOD is defined as the sum of all trace gases oxidation rate by OH, NO3, and O3. Is NO included? Please declare it clearly. 5. Figure 10. It's good to have one more role showing the integral oxidation over a day. 6. Figure 11. Why alkenes show a significant variation in RNO3 and RO3 but not ROH? 7. Figure 12. Maybe it's better to use the same scale for all panels.

---

## Referee Comment (RC3) · Anonymous Referee #1 · 11 Jan 2020

The manuscript Parameterized reactivity of hydroxy radical, ozone, nitrate radical and atmospheric oxidation capacity during summer at a suburban site between Beijing and Tianjin by Yang et al. describes atmospheric reactivities towards OH, ozone and nitrate of several trace gases and their oxidation capacities from measurements of trace gases conducted during one-month of field campaign in Xiang he during summer 2018. The authors use an extensive dataset of concentrations of trace gases, including $O_3$, $NO_x$, $NO_y$, $SO_2$, $CO$ and VOCs, with meteorological parameters and photolysis frequencies for calculating OH, $O_3$ and $NO_3$ reactivities as well as their atmospheric

oxidation capacities and describe the current air chemistry over the region during summertime when photochemistry is enhanced. I find the manuscript interesting in the way it addresses the air chemistry regime over a sensitive highly polluted region and suggests how to implement current environmental policies for improving the quality of air. I would have found the manuscript more accurate if OH/NO3 reactivities could be measured along with the trace gases during the campaign. Calculated reactivities need the associated uncertainty (from the measurements and from the reactions constants). Additionally, the parametrization used to determine the oxidation capacity needs better description and the associated uncertainty. Nevertheless, I find the manuscript suitable for the journal ACP and I recommend its publication after some changes will be considered.

General comment:

I suggest to include a short comment in the discussion of the results considering the missing reactivity fractions reported in highly polluted urban/suburban environments and studies in China, where available. This could possibly lead to different (more pessimistic or optimistic) scenarios than the one reported in the present study that is worth knowing to the reader. The manuscript is sometimes not very fluent either for the intensive use of acronyms or language phrasing, making the reading at times a bit complicated. I suggested some rephrasing but you might want to improve the fluency by making some concepts more concise and use a simplified nomenclature. I also suggest to revise the length of the abstract, of keywords used, number of figures and some parts of the discussion. You might also want to reconsider the title for a shorter one (for example, something like: parameterized atmospheric reactivity and oxidation capacity during summer . . .).

Specific comments:

p. 2 L 26 "that result" p. 2 L 27 "the air chemistry" instead of self-cleansing capacity p. 2 L 30 which network? Specify. Avoid references in the abstract as the personal

communication. p.2 L 35 use 48-99% p.2 L 36-40 try rephrasing with less acronyms p.2 L 40-43 give less details as the calculation is not yet explained p.2 L 43-47 Leave out this information p.3 L 47-49 Keep this information p.3 L 49-51 For conclusions p.3 L 52 suggested keywords: VOCs, atmospheric oxidants reactivity, atmospheric oxidizing capacity, North China Plain p.4 L 70 give an estimate p.4 L 72 remove a major part p.4 L 72 by reactions with atmospheric oxidants p.4 L 80 comparative reactivity method cit. Sinha et al., 2008 p.5 L 93 you can add the study of Helsinki (Praplan et al.) and of Seoul (Kim et al.) p.5 L 100 you can add the study done in the PO valley (Kaiser et al., 2015) and in India (Kumar et al., 2018) p.5 L 101 the range will change with the measurements done in India p.5 L 106 metric instead of matric, check also other parts of the manuscript p.6 L 112 You can cite the study of Mogensen et al. p.6 L 114 "as reported.." p.6 L 114 contribution from NOx p.6 L 117 Does this comparison point at the use of different fuels/vehicles used? p.6 L 123 due to NO3 elevated concentrations at night p.7 L 141 of the reactions for some alkenes. . . p.7 L 146 BERLIOZ and NOTOMO/ before write the type of environment and where then you can add in brackets the name of the campaign. p.7 L 151-152 remove sentence p.8 L 156 In this study, we calculated the OH, O3 and NO3 reactivities from VOCs measurements . . . p.8 L 158 we calculated the oxidation capacities of xx xx xx and estimated their relative contributions. p.8 L 175 detection limit instead of lowest detection limit, please modify also where else is mentioned. p.9 L 190 Please refer to Wang et al., (2014b) for more details about the techniques used. p.9 L 193 remove samples p.9 L 193 Is the GC system having 2 columns or columns were exchanged on different campaign periods? Please specify p.9 L 193 How was the sampling conducted? Which type of inlet was used? Was there any O3 scrubber used to measure alkenes? In general, are the VOC measurements and atmospheric events from this campaign described elsewhere? p.12 L 243 Please specify how close the sensors were to the measurement area. p.12 L 247 The nomenclature of radical reactivity where O3 is considered is incorrect. Please change this word where used in the text with "atmospheric oxidants reactivity" or something similar that can commonly include OH, NO3 and O3. p.12 L 248-253 Needs rephrasing. You

can express the same concept with one sentence, for example: atmospheric oxidants reactivity is a measure of the strength of reaction of trace gases to the three main atmospheric oxidants. . . You can cite the first studies that introduced this concept (check for Brune et al., or Kovacs and Brune) and remove the references not needed here. p.12 L 247&274 You can include a table with all rate coefficients used and respective references for these 2 sections p.16 L 331 Define OVOC p.16 L 335 I suggest to avoid the use of many acronyms when is not extremely needed, TVOC can be written as total VOC. There are many acronyms in the manuscript and the reader is sometimes lost. p.16 L 348 for secondary species p.16 L 351-352 remove lines p.23 L 503-505 You can shorten the discussion by removing from which are 3 etc.. p.25 L 543-555 Please make the concept more concise and present it in the methods part p.26 L 569 This is an interesting section. Can you implement the discussion by indicating the sources of the VOCs whose concentration could be limited and make some concrete examples for the region under study? p.30 L 644 what do you mean exactly by integral of the oxidation rate? This concept needs to be clarified. Can you (briefly) illustrate the 2 type of concepts of the oxidation rate results in the method section? Same for what you are illustrating in figures 10 &13. Also it is confusing using both approaches, you might want to make a table with the results from the 2 approaches and discuss the differences rather than discuss the two of them separately, it will make the discussion part also more clear. p.31 L 682-683 is overestimated Figures Fig 2. Move the legends of the panels out of the graphs. Add minor ticks on the left /right axes Fig. 3 where is NO2 in the right panel? Fig. 5 include a table clarifying which are the BVOC considered and OVOC considered Fig.10 Unsaturated VOC: there should be a larger contribution during daytime given by O3, why this is not the case? 13 figures are many. You might want to simplify the manuscript keeping only the most relevant ones in the main body and leave the others to the supplementary information (I suggest to keep 1, 2, 3, 4, (5 could be presented as a table instead of graphically), 6 or 7, 8& 9) Table 1& Table S1. Please readapt these tables to a table/ tables where: concentration, SD, reactivities, reaction coeffs, and refs are included. If the table is too big you can split it in two tables

(concentration, SD, reactivities) and reaction coefficients and references. The chemicals should be grouped according to the nomenclature used in the manuscript (BVOC, OVOC. . .etc) Supplementary material: Please include some explanations between the figures.

---

## Author Comment (AC1) · 27 Apr 2020

We thank the reviewers for the constructive comments and suggestions, which are very helpful to improve scientific content of the manuscript. We have revised the manuscript accordingly and addressed all the reviewers' comments point-by-point for consideration as below. The remarks from the reviewers are shown in black, and our responses are shown in blue color. All the page and line numbers mentioned following are refer to the revised manuscript without change tracked.

**Reviewer #1**

The manuscript Parameterized reactivity of hydroxy radical, ozone, nitrate radical and atmospheric oxidation capacity during summer at a suburban site between Beijing and Tianjin by Yang et al. describes atmospheric reactivities towards OH, ozone and nitrate of several trace gases and their oxidation capacities from measurements of trace gases conducted during one-month of field campaign in Xiang he during summer 2018. The authors use an extensive dataset of concentrations of trace gases, including $O_3$, $NO_x$, $NO_y$, $SO_2$, CO and VOCs, with meteorological parameters and photolysis frequencies for calculating OH, $O_3$ and $NO_3$ reactivities as well as their atmospheric oxidation capacities and describe the current air chemistry over the region during summertime when photochemistry is enhanced. I find the manuscript interesting in the way it addresses the air chemistry regime over a sensitive highly polluted region and suggests how to implement current environmental policies for improving the quality of air. I would have found the manuscript more accurate if OH/$NO_3$ reactivities could be measured along with the trace gases during the campaign. Calculated reactivities need the associated uncertainty (from the measurements and from the reactions constants). Additionally, the parametrization used to determine the oxidation capacity needs better description and the associated uncertainty. Nevertheless, I find the manuscript suitable for the journal ACP and I recommend its publication after some changes will be considered.

General comment:

I suggest to include a short comment in the discussion of the results considering the missing reactivity fractions reported in highly polluted urban/suburban environments and studies in China, where available. This could possibly lead to different (more pessimistic or optimistic) scenarios than the one reported in the present study that is worth knowing to the reader. The manuscript is sometimes not very fluent either for the intensive use of acronyms or language phrasing, making the reading at times a bit complicated. I suggested some rephrasing but you might want to improve the fluency by making some concepts more concise and use a simplified nomenclature. I also suggest to revise the length of the abstract, of keywords used, number of figures and some parts of the discussion. You might also want to reconsider the title for a shorter one (for example, something like: parameterized atmospheric reactivity and oxidation capacity during summer :).

Response: we thank the reviewer for the positive comments. We added discussions about the missing reactivity reported in polluted environment in China (Line 384-391 in the revised version). Also, the manuscript has been shortened considerably and the title has been changed according to the suggestion.

Specific comments:

p.2 L26 "that result"

Response: Thanks for the suggestion. We have corrected the 'resulted' to 'result'. Please refer to Line 32 in the revised version.

p. 2 L27 "the air chemistry" instead of self-cleansing capacity

Response: The 'self-cleansing capacity' have been corrected to 'the air chemistry'. Please refer to Line 33 in the revised version.

p.2 L30 which network? Specify. Avoid references in the abstract as the personal communication.

Response: Thanks for the suggestion. Relying on the Campaign on Atmospheric Aerosol Research network of China

(CARE-China) launched by the Chinese Academy of Sciences (CAS) in 2011 (Xin et al., 2015), field VOC samples were collected simultaneously at 29 sites across China from 2012 to 2014. In order to avoid references in the abstract as the personal communication, we decided to delete the statement 'The site had suffered the most abundant annual mean VOCs concentrations according to a network observation from 2012-2014 (personal communication)'.

**Reference:**

Xin, J. Y., Wang, Y. S., Pan, Y. P., Ji, D. S., Liu, Z. R., Wen, T. X., Wang, Y. H., Li, X. R., Sun, Y., Sun, J., Wang, P. C., Wang, G. H., Wang, X. M., Cong, Z. Y., Song, T., Hu, B., Wang, L. L., Tang, G. Q., Gao, W. K., Guo, Y. H., Miao, H. Y., Tian, S. L., and Wang, L.: The Campaign on Atmospheric Aerosol Research Network of China Care-China, B Am Meteorol Soc, 96, 1137-1155, do i:10.1175/Bams-D-14-00039.1, 2015.

p.2 L35 use 48-99%

Response: Thanks for the suggestion. 47, 99 and 99% were used instead of 43-99%. Please refer to Line 39 in the revised version.

p.2 L 36-40 try rephrasing with less acronyms

Response: We have rewritten the sentence as follows:

Alkenes dominated the OH, $NO_3$ and $O_3$ reactivities towards total non-methane volatile organic compounds (NMVOCs), accounting for 42.9%, 77.8% and 94.0%, respectively. The total OH, $NO_3$ and $O_3$ reactivities displayed a similar diurnal variation with the lowest during the afternoon but the highest during the rush hours, and the diurnal profile of NOx appears to be the major driver for the diurnal profiles of the three oxidant reactivities. Please refer to Line 39-43 in the revised version.

p.2 L40-43 give less details as the calculation is not yet explained

Response: Thanks for the suggestion. We have explained the calculation. The AOC was confirmed by quantifying the loss rates of NMVOCs, $CH_4$ and CO via reactions with OH, $O_3$ and $NO_3$ (Line 295-301 in the revised version).

p.2 L43-47 Leave out this information

Response: We have left out this information.

p.3 L47-49 Keep this information

Response: Thanks for the suggestion. We have kept this information. Please refer to Line 49-50 in the revised version.

p.3 L49-51 For conclusions

Response: Thanks for the suggestion. we revised this sentence as 'We suggest that further studies, especially direct observations of OH and $NO_3$ radicals concentrations and their reactivities, are required to better understanding the trace gas reactivity and AOC.' Please refer to Line 50-52 in the revised version.

p.3 L52 suggested keywords: VOCs, atmospheric oxidants reactivity, atmospheric oxidizing

capacity, North China Plain

Response: We have followed the comments and the keywords has been corrected to 'VOCs, atmospheric oxidants reactivity, atmospheric oxidation capacity, North China Plain'. Please refer to Line 54 in the revised version.

p.4 L70 give an estimate

Response: Thanks for the suggestion. We cannot give an estimate. So, we have restructured this sentence as follows: In the planetary boundary layer, high concentrations of primary pollutants, such as carbon monoxide (CO), nitrogen oxides (NOx=NO+$NO_2$) and volatile organic compounds (VOCs) from both biogenic and anthropogenic origins, are transformed by reactions with atmospheric oxidants, such as hydroxyl (OH) radicals, nitrate ($NO_3$) radicals, chlorine atom and ozone ($O_3$) on local to global scales (Atkinson and Arey, 2003;Heard and Pilling, 2003;Lu et al., 2018), with the dominant reaction depending on the time of day and specific trace gases. Please refer to Line 57-62 in the revised version.

**References:**

Atkinson, R., and Arey, J.: Atmospheric Degradation of Volatile Organic Compounds., Chemical Reviews, 103,

4605-4638, doi:10.102/cr0206420, 2003.

Heard, D. E., and Pilling, M. J.: Measurement of OH and HO$_2$ in the troposphere, Chemical Reviews, 103, 5163-5198, doi:10.1021/cr020522s, 2003.

Lu, K., Guo, S., Tan, Z., Wang, H., Shang, D., Liu, Y., Li, X., Wu, Z., Hu, M., and Zhang, Y.: Exploring the Atmospheric Free Radical chemistry in China: The Self-Cleansing Capacity and the Formation of Secondary air Pollution, National Science Review, doi:10.1093/nsr/nwy073, 2018.

p.4 L72 remove a major part

Response: We have removed 'a major part'. Please refer to Line 59 in the revised version.

p.4 L72 by reactions with atmospheric oxidants

Response: The 'by reactions with free radicals' have been replaced with 'by reactions with atmospheric oxidants.' Please refer to Line 59 in the revised version.

p.4 L80 comparative reactivity method cit. Sinha et al., 2008

Response: Thanks for the suggestion. Sinha et al., 2008 has been cited after "… a comparative rate method" as follows: The online techniques used to determine OH reactivity include the flow tube with sliding injector method (Kovacs et al., 2003), a comparative rate method (Sinha et al., 2008) and a laser flash photolysis pump probe technique (Whalley et al., 2016). Please refer to Line 72-75 in the revised version.

**References:**

Kovacs, T. A., Brune, W. H., Harder, H., Martinez, M., Simpas, J. B., Frost, G. J., Williams, E., Jobson, T., Stroud, C., Young, V., Fried, A., and Wert, B.: Direct measurements of urban OH reactivity during Nashville SOS in summer 1999, J Environ Monitor, 5, 68-74, doi:10.1039/b204339d, 2003.

Sinha, V., Williams, J., Crowley, J. N., and Lelieveld, J.: The Comparative Reactivity Method – a new tool to measure total OH Reactivity in ambient air, Atmos. Chem. Phys., 8, 2213-2227, doi:10.5194/acp-8-2213-2008, 2008.

Whalley, L. K., Stone, D., Bandy, B., Dunmore, R., Hamilton, J. F., Hopkins, J., Lee, J. D., Lewis, A. C., and Heard, D. E.: Atmospheric OH reactivity in central London: observations, model predictions and estimates of in situ ozone production, Atmos Chem Phys, 16, 2109-2122, doi:10.5194/acp-16-2109-2016, 2016.

p.5 L93 you can add the study of Helsinki (Praplan et al.) and of Seoul (Kim et al.)

Response: We added the study of Helsinki (Praplan et al.) and of Seoul (Kim et al.) as follows:

The urban areas investigated included Nashville, USA (SOS) (Kovacs et al., 2003), New York, USA (PMTACS-NY2004) (Ren et al., 2006a), Mexico City, Mexico (MCMA-2003) (Shirley et al., 2006), Houston, USA (TRAMP2006) (Mao et al., 2010), Paris, France (MEGAPOLI) (Dolgorouky et al., 2012), London, UK (ClearfLo) (Whalley et al., 2016), Helsinki, Finland (Praplan et al., 2017), Seoul, South Korea (Kim et al., 2016) and Beijing, China (Yang et al., 2017). Please refer to Line 76-81 in the revised version.

**References:**

Dolgorouky, C., Gros, V., Sarda-Esteve, R., Sinha, V., Williams, J., Marchand, N., Sauvage, S., Poulain, L., Sciare, J., and Bonsang, B.: Total OH reactivity measurements in Paris during the 2010 MEGAPOLI winter campaign, Atmos Chem Phys, 12, 9593-9612, doi:10.5194/acp-12-9593-2012, 2012.

Kim, S., Sanchez, D., Wang, M., Seco, R., Jeong, D., Hughes, S., Barletta, B., Blake, D. R., Jung, J., Kim, D., Lee, G., Lee, M., Ahn, J., Lee, S. D., Cho, G., Sung, M. Y., Lee, Y. H., Kim, D. B., Kim, Y., Woo, J. H., Jo, D., Park, R., Park, J. H., Hong, Y. D., and Hong, J. H.: OH reactivity in urban and suburban regions in Seoul, South Korea - an East Asian megacity in a rapid transition, Faraday Discuss, 189, 231-251, doi:10.1039/c5fd00230c, 2016.

Kovacs, T. A., Brune, W. H., Harder, H., Martinez, M., Simpas, J. B., Frost, G. J., Williams, E., Jobson, T., Stroud, C., Young, V., Fried, A., and Wert, B.: Direct measurements of urban OH reactivity during Nashville SOS in summer 1999, J Environ Monitor, 5, 68-74, doi:10.1039/b204339d, 2003.

Mao, J., Ren, X., Chen, S., Brune, W. H., Chen, Z., Martinez, M., Harder, H., Lefer, B., Rappenglück, B., Flynn, J.,

and Leuchner, M.: Atmospheric oxidation capacity in the summer of Houston 2006: Comparison with summer measurements in other metropolitan studies, Atmos Environ, 44, 4107-4115, doi:10.1016/j.atmosenv.2009.01.013, 2010.

Praplan, A. P., Pfannerstill, E. Y., Williams, J., and Hellén, H.: OH reactivity of the urban air in Helsinki, Finland, during winter, Atmos Environ, 169, 150-161, doi:10.1016/j.atmosenv.2017.09.013, 2017.

Ren, X., Brune, W. H., Mao, J., Mitchell, M. J., Lesher, R. L., Simpas, J. B., Metcalf, A. R., Schwab, J. J., Cai, C., and Li, Y.: Behavior of OH and HO$_2$ in the winter atmosphere in New York City, Atmos Environ, 40, 252-263, doi:10.1016/j.atmosenv.2005.11.073, 2006.

Shirley, T. R., Brune, W. H., Ren, X., Mao, J., Lesher, R., Cardenas, B., Volkamer, R., Molina, L. T., Molina, M. J., Lamb, B., Velasco, E., Jobson, T., and Alexander, M.: Atmospheric oxidation in the Mexico City Metropolitan Area (MCMA) during April 2003, Atmos Chem Phys, 6, 2753-2765, doi:10.5194/acp-6-2753-2006, 2006.

Whalley, L. K., Stone, D., Bandy, B., Dunmore, R., Hamilton, J. F., Hopkins, J., Lee, J. D., Lewis, A. C., and Heard, D. E.: Atmospheric OH reactivity in central London: observations, model predictions and estimates of in situ ozone production, Atmos Chem Phys, 16, 2109-2122, doi:10.5194/acp-16-2109-2016, 2016.

Yang, Y., Shao, M., Keßel, S., Li, Y., Lu, K., Lu, S., Williams, J., Zhang, Y., Zeng, L., Nölscher, A. C., Wu, Y., Wang, X., and Zheng, J.: How the OH reactivity affects the ozone production efficiency: case studies in Beijing and Heshan, China, Atmos Chem Phys, 17, 7127-7142, doi:10.5194/acp-17-7127-2017, 2017.

p.5 L100 you can add the study done in the PO valley (Kaiser et al., 2015) and in India (Kumar et al., 2018)

Response: Thanks for the suggestion. We have added the study done in the PO valley (Kaiser et al., 2015) and in India (Kumar et al., 2018) as follows:The suburban areas investigated included Whiteface Mountain, USA (PMTACS-NY2002) (Ren et al., 2006), Weybourne, UK (TORCH-2) (Lee et al., 2010), Yufa, China (CAREBeijing-2006) (Lu et al., 2010), Backgarden, China (PRIDE-PRD) (Lou et al., 2010), Jülich, Germany (HOx Comp) (Elshorbany et al., 2012), Ersa, Corsica (CARBOSOR-ChArMeX) (Zannoni et al., 2017), Po Valley, Italy (Kaiser et al., 2015), Indo-Gangetic Plain, India (Kumar et al., 2018) and Heshan, China (Yang et al., 2017). Please refer to Line 84-89 in the revised version.

Response: The GC system have 2 columns. Column 1 is a PLOT-Al$_2$O$_3$ column (15 m × 0.32 mm ID×3 μm, J&W Scientific, USA) separating C2-C5 hydrocarbons and then measured by the FID; Column 2 is a semi polar column (DB624, 60 m × 0.25 mm ID×1.4 μm, J&W Scientific, USA) separating other compounds and then quantified using a quadrupole MS detector. The two columns were not exchanged during the intensive measurement campaign from 6 July 2018 to 6 August 2018.Please refer to Line 199-205 in the revised version.

p.9 L193 How was the sampling conducted? Which type of inlet was used? Was there any O3 scrubber used to measure alkenes? In general, are the VOC measurements and atmospheric events from this campaign described

elsewhere?

Response: Briefly, Samples are collected into GC-MS/FID at a flow rate of 60 mL min$^{-1}$ with sampling time of 5 min at the beginning of each hour. The sampling lines for ambient air and standard gases were both Teflon tubes with a 1/4-inch outside diameter (OD). A Teflon filter was placed in the inlet to prevent particulate matters from entering the instrument, and a water trap was used to remove $H_2O$ from the air samples. Ascarite II was used to remove $CO_2$ and $O_3$ before the FID channel, whereas a $Na_2SO_3$ trap was used to remove $O_3$ in the MS channel. Please refer to Line 194-205 in the revised version. The NMVOCs measurements and atmospheric events from this campaign are not described elsewhere.

p.12 L243 Please specify how close the sensors were to the measurement area.

Response: Thanks for the suggestion. The sensors are about 3000 meters away from the measurement area. Please refer to Line 243 in the revised version.

p.12 L247 the nomenclature of radical reactivity where $O_3$ is considered is incorrect. Please change this word where used in the text with "atmospheric oxidants reactivity" or something similar that can commonly include OH, NO3 and O3.

Response: Thanks for the correction. Speciated oxidant reactivity was used instead of speciated radical reactivity. Please refer to Line 274 in the revised version.

p.12 L248-253 Needs rephrasing. You can express the same concept with one sentence, for example: atmospheric oxidants reactivity is a measure of the strength of reaction of trace gases to the three main atmospheric oxidants: : You can cite the first studies that introduced this concept (check for Brune et al., or Kovacs and Brune) and remove the references not needed here.

Response: We have followed the comments and expressed the same concept with one sentence as follows: Atmospheric oxidant reactivity is a measure of the strength of reaction of trace gases to the oxidant (= OH, $O_3$, $NO_3$) (Kovacs et al., 2003;Mogensen et al., 2015). High oxidant reactivity values correspond to short lifetimes and long-lived species have low reactivities. The total OH, $NO_3$ and $O_3$ reactivities can be calculated by Eq. (1)-(3), respectively. Please refer to Line 275-278 in the revised version.

Response: Thanks for the suggestion. We have added a table with all temperature-dependent reaction rate coefficients used and respective references, and listed them in Table S1 in supplement information as follows:

Table S1. The temperature-dependent reaction rate coefficients of trace gases with OH radical, $O_3$ and $NO_3$ radical used in this study.

| Species | Temperature-dependence of $k_{OH}$ (cm$^3$ molecule$^{-1}$ s$^{-1}$) | Temperature-dependence of $k_{O3}$ (cm$^3$ molecule$^{-1}$ s$^{-1}$) | Temperature-dependence of $k_{NO3}$ (cm$^3$ molecule$^{-1}$ s$^{-1}$) |
|---|---|---|---|
| CH$_4$ | $1.85 \times 10^{-12}$exp(-1690/T) | $<1 \times 10^{-23}$ | $<1 \times 10^{-18}$ |
| Alkanes | | | |
| Ethane | $6.9 \times 10^{-12}$exp(-1000/T) | $<1 \times 10^{-23}$ | $<1 \times 10^{-17}$ |
| Propane | $7.6 \times 10^{-12}$exp(-585/T)×0.736 | $<1 \times 10^{-23}$ | $<7 \times 10^{-17}$ |

| | | | |
|---|---|---|---|
| iso-Butane | $1.16\times10^{-17}\times T^2\times exp(225/T)\times0.794$ | $<1\times10^{-23}$ | $1.06\times10^{-16}$ |
| n-Butane | $9.8\times10^{-12}exp(-425/T)\times0.873$ | $<1\times10^{-23}$ | $2.8\times10^{-12}exp(-3280/T)$ |
| Cyclopentane | $4.97\times10^{-12}$ | $<1\times10^{-23}$ | $1.4\times10^{-16}$ |
| iso-Pentane | $3.6\times10^{-12}$ | $<1\times10^{-23}$ | $1.62\times10^{-16}$ |
| n-Pentane | $2.44\times10^{-17}\times T^2\times exp(183/T)\times0.568$ | $<1\times10^{-23}$ | $8.7\times10^{-17}$ |
| 2,2-Dimethylbutane | $3.22\times10^{-11}exp\ (-781/T)\times0.632$ | $<1\times10^{-23}$ | $4.4\times10^{-16}$ |
| 2,3-Dimethylbutane | $1.24\times10^{-17}\times T^2\times exp(494/T)\times0.877$ | $<1\times10^{-23}$ | $4.4\times10^{-16}$ |
| 2-Methylpentane | $5.4\times10^{-12}$ | $<1\times10^{-23}$ | $1.8\times10^{-16}$ |
| 3-Methylpentane | $5.2\times10^{-12}$ | $<1\times10^{-23}$ | $2.2\times10^{-16}$ |
| n-Hexane | $1.53\times10^{-17}\times T^2\times exp(414/T)\times0.061$ | $<1\times10^{-23}$ | $1.1\times10^{-16}$ |
| 2,4-Dimethylpentane | $4.77\times10^{-12}$ | $<1\times10^{-23}$ | $1.5\times10^{-16}$ |
| Methylcyclopentane | $5.2\times10^{-12}$ | $<1\times10^{-23}$ | $1.4\times10^{-16}$ |
| 2-Methylhexane | $5.65\times10^{-12}$ | $<1\times10^{-23}$ | $1.5\times10^{-16}$ |
| 2,3-Dimethylpentane | $1.5\times10^{-12}$ | $<1\times10^{-23}$ | $1.5\times10^{-16}$ |
| Cyclohexane | $2.88\times10^{-17}exp(309/T)$ | $<1\times10^{-23}$ | $1.4\times10^{-16}$ |
| 3-Methylhexane | $5.6\times10^{-12}$ | $<1\times10^{-23}$ | $1.5\times10^{-16}$ |
| 2,2,4-Trimethylpentane | $3.34\times10^{-12}$ | $<1\times10^{-23}$ | $9.0\times10^{-17}$ |
| n-Heptane | $1.59\times10^{-17}\times T^2\times exp(478/T)$ | $<1\times10^{-23}$ | $1.5\times10^{-16}$ |
| Methylcyclohexane | $4.97\times10^{-12}$ | $<1\times10^{-23}$ | $1.4\times10^{-16}$ |
| 2,3,4-Trimethylpentane | $6.6\times10^{-12}$ | $<1\times10^{-23}$ | $1.9\times10^{-16}$ |
| 2-Methylheptane | $7\times10^{-12}$ | $<1\times10^{-23}$ | $1.9\times10^{-16}$ |
| 3-Methylheptane | $7\times10^{-12}$ | $<1\times10^{-23}$ | $1.9\times10^{-16}$ |
| n-Octane | $2.76\times10^{-17}\times T^2\times exp(378/T)$ | $<1\times10^{-23}$ | $1.9\times10^{-16}$ |
| Nonane | $2.51\times10^{-17}\times T^2\times exp(477/T)$ | $<1\times10^{-23}$ | $2.3\times10^{-16}$ |
| n-Decane | $3.13\times10^{-17}\times T^2\times exp(416/T)$ | $<1\times10^{-23}$ | $2.8\times10^{-16}$ |
| n-Undecane | $12.3\times10^{-12}$ | $<1\times10^{-23}$ | |
| Alkenes | | | |
| Ethylene | $9.0\times10^{-12}\ (T/300)^{-0.85}$ | $9.1\times10^{-15}exp(-2580/T)$ | $3.3\times10^{-12}exp(-2880/T)$ |
| Propylene | $3.0\times10^{-11}(T/300)^{-1}$ | $5.5\times10^{-15}exp(-1880/T)$ | $4.6\times10^{-13}exp(-1155/T)$ |
| trans-2-Butene | $1.01\times10^{-11}exp\ (550/T)$ | $6.64\times10^{-15}exp(-1095/T)$ | $3.9\times10^{-13}$ |
| 1-Butene | $6.6\times10^{-12}\ exp(465/T)\times0.87$ | $9.64\times10^{-18}$ | $1.35\times10^{-14}$ |
| cis-2-Butene | $1.1\times10^{-11}\ exp(487/T)$ | $3.22\times10^{-15}exp(-968/T)$ | $3.52\times10^{-13}$ |
| 1,3-Butadiene | $1.48\times10^{-11}exp(448/T)\times0.649$ | $1.34\times10^{-14}exp(-2283/T)\times0.5$ | $1.0\times10^{-13}$ |
| 1-Pentene | $5.86\times10^{-12}\ exp(500/T)\times0.87$ | $1.06\times10^{-17}$ | $1.5\times10^{-14}$ |
| trans-2-Pentene | $6.7\times10^{-11}$ | $1.6\times10^{-16}$ | $3.7\times10^{-13}$ |
| cis-2-Pentene | $6.5\times10^{-11}$ | $1.3\times10^{-16}$ | $3.7\times10^{-13}$ |
| Isoprene | $2.7\times10^{-11}exp(390/T)$ | $1.03\times10^{-14}exp(-1995/T)$ | $3.15\times10^{-12}exp(-450/T)$ |
| 1-Hexene | $3.7\times10^{-11}$ | $1.31\times10^{-17}$ | $1.8\times10^{-14}$ |
| OVOCs | | | |
| HCHO | $5.4\times10^{-12}exp(135/T)$ | $<1\times10^{-20}$ | $5.6\times10^{-16}$ |
| Acrolein | $18.3$ | $<1\times10^{-20}$ | |
| Propanal | $5.1\times10^{-12}exp(405/T)$ | $<1\times10^{-20}$ | $6.4\times10^{-15}$ |
| Acetone | $8.8\times10^{-12}exp(-1320/T)+$ | $<1\times10^{-20}$ | $<3\times10^{-17}$ |

| | | | |
|---|---|---|---|
| | $1.7\times10^{-14}\exp(423/T)$ | | |
| MTBE | $2.94\times10^{-12}$ | $<1\times10^{-20}$ | |
| Methacrolein | $8.0\times10^{-12}\exp(380/T)$ | $1.4\times10^{-15}\exp(-2100/T)$ | $3.4\times10^{-15}$ |
| n-Butanal | $6.0\times10^{-12}\exp(410/T)$ | $<1\times10^{-20}$ | $1.7\times10^{-12}\exp(-1500/T)$ |
| MethylVinylKetone | $2.6\times10^{-12}\exp(610/T)$ | $<1\times10^{-20}$ | $6.0\times10^{-16}$ |
| Methylethylketone | $1.5\times10^{-12}\exp(-90/T)\times0.462$ | $<1\times10^{-20}$ | |
| 2-Pentanone | $4.4\times10^{-12}$ | $<1\times10^{-20}$ | |
| Pentanal | $6.34\times10^{-12}\exp(448/T)\times0.19$ | $<1\times10^{-20}$ | $1.5\times10^{-14}$ |
| 3-Pentanone | $2\times10^{-12}$ | $<1\times10^{-20}$ | |
| Hexanal | $3.0\times10^{-11}$ | $<1\times10^{-20}$ | $1.6\times10^{-14}$ |
| Aromatics | | | |
| Benzene | $2.3\times10^{-12}\exp(-190/T)\times0.53$ | $<1\times10^{-20}$ | $3.0\times10^{-17}$ |
| Toluene | $1.8\times10^{-12}\exp(340/T)\times0.18$ | $<1\times10^{-20}$ | $7.0\times10^{-17}$ |
| Ethylbenzene | $7\times10^{-12}$ | $<1\times10^{-20}$ | $6.0\times10^{-16}$ |
| m/p-Xylene | $1.89\times10^{-11}$ | $<1\times10^{-20}$ | $2.6\times10^{-16}$ |
| o-Xylene | $1.36\times10^{-11}$ | $<1\times10^{-20}$ | $4.1\times10^{-16}$ |
| Styrene | $5.8\times10^{-11}$ | $1.7\times10^{-17}$ | $1.5\times10^{-12}$ |
| Isopropylbenzene | $6.3\times10^{-12}$ | $<1\times10^{-20}$ | $6.0\times10^{-16}$ |
| n-Propylbenzene | $5.8\times10^{-12}$ | $<1\times10^{-20}$ | $6.0\times10^{-16}$ |
| m-Ethyltoluene | $1.18\times10^{-11}$ | $<1\times10^{-20}$ | $8.6\times10^{-16}$ |
| p-Ethyltoluene | $1.86\times10^{-11}$ | $<1\times10^{-20}$ | $8.6\times10^{-16}$ |
| 1,3,5-Trimethylbenzene | $5.67\times10^{-11}$ | $<1\times10^{-20}$ | $8.8\times10^{-16}$ |
| o-Ethyltoluene | $1.19\times10^{-11}$ | $<1\times10^{-20}$ | $8.6\times10^{-16}$ |
| 1,2,4-Trimethylbenzene | $3.25\times10^{-11}$ | $<1\times10^{-20}$ | $1.8\times10^{-15}$ |
| 1,2,3-Trimethylbenzene | $3.27\times10^{-11}$ | $<1\times10^{-20}$ | $1.9\times10^{-15}$ |
| Criteria pollutants | | | |
| CO | $2.4\times10^{-13}$ | | |
| NO | $3.3\times10^{-11}(T/300)^{-0.3}$ | $1.4\times10^{-12}\exp(-1310/T)$ | $1.8\times10^{-11}\exp(110/T)$ |
| NO$_2$ | $4.1\times10^{-11}$ | $1.4\times10^{-13}\exp(-2470/T)$ | $1.9\times10^{-12}(T/300)^{0.2}$ |
| SO$_2$ | $1.3\times10^{-12}(T/300)^{-0.7}$ | | $<1.0\times10^{-19}$ |
| O$_3$ | $1.7\times10^{-12}\exp(-940/T)$ | | |

Note: The temperature-dependent reaction rate coefficients of VOCs and CO are from Atkinson et al. (1983), Atkinson and Arey (2003), Atkinson et al. (2006), Salgado et al. (2008) and the Master Chemical Mechanism, MCM v3.3.1 via the website: http://mcm.leeds.ac.uk/MCM (last accessed: 25 March 2020); The temperature-dependent reaction rate coefficients of NO, NO$_2$, SO$_2$ and O$_3$ are from Atkinson et al. (2004). T denotes temperature.

p.26 L569 This is an interesting section. Can you implement the discussion by indicating the sources of the VOCs whose concentration could be limited and make some concrete examples for the region under study?

Response: Thanks for the suggestion. We agreed that identifying the possible sources of the VOCs whose concentration could be limited is important to provide recommendations for future policies. So, the origin of key species was initially identified based on the certain chemical tracers which are generally presumed to be emitted from specific sources and present in significant amounts in the collected samples. Please refer to Line 567-570 and 584-588 in the revised version.

p.30 L644 what do you mean exactly by integral of the oxidation rate? This concept needs to be clarified. Can you (briefly) illustrate the 2 type of concepts of the oxidation rate results in the method section? Same for what you are illustrating in figures 10 &13. Also, it is confusing using both approaches, you might want to make a table with the results from the 2 approaches and discuss the differences rather than discuss the two of them separately, it will make the discussion part also clearer.

Response: Thanks for the suggestion. We are sorry for this confusing statement of integral of the oxidation rate in the previous manuscript, and 'averaged loss rates' was used instead of 'integral of the oxidation rate' in the revised version. Please refer to Line 662 in the revised version.

The concept of the oxidation rate is same as AOC. The Figure 10 in the previous manuscript (Figure 9 in the revised version) shows the overall loss rate of NMVOCs, $CH_4$ and CO via reactions with OH, $O_3$ and $NO_3$, but Figure 13 in the previous manuscript (Figure S17 in the revised version) shows the loss rate of NMVOCs groups, illustrating the relative importance of speciated NMVOCs oxidation pathways.

p.31 L682-683 is overestimated

Response: We have removed the sentence in that we have simulated the OH and $NO_3$ mixing ratios using atmospheric chemistry transport model SOSAA.

Figures

Fig 2. Move the legends of the panels out of the graphs. Add minor ticks on the left /right axes

Response: Thanks for the suggestion. We have moved the legends of the panels out of the graphs and added minor ticks on the left /right axes.

[Figure]

Figure 2. Time series of meteorology parameters, trace gases, photolysis rates and VOCs concentrations during the field campaign at Xianghe from 6 July to 6 August 2018.

Fig. 3 where is NO2 in the right panel?

Response: Thanks for the suggestion. We have redrawn the Fig. 3 focusing on the NMVOCs.

[Figure]

Figure 3. The top 10 NMVOCs' contribution to VOCs concentration (a), OH reactivity (b), NO₃ reactivity (c) and O₃ reactivity (d)during the field campaign at Xianghe from 6 July to 6 August 2018.

Fig. 5 include a table clarifying which are the BVOC considered and OVOC considered

Response: We have added a table clarifying VOCs groups and species included, and listed them in Table S1 in supplement information.

Fig.10 Unsaturated VOC: there should be a larger contribution during daytime given by O3, why this is not the case?

Response: In terms of alkenes, O₃ indeed make a larger contribution during daytime. This can be accounted for by the following facts: the alkenes reaction rate coefficients with O₃ are much higher than alkanes, aromatics and OVOCs reaction rate coefficients with O₃; 2) the orders of magnitude of the differences of alkenes reaction rate coefficients with OH, O₃ and NO₃ are much smaller than that alkanes, aromatics and OVOCs reaction rate coefficients with OH, O₃ and NO₃. However, in this study, unsaturated VOCs including cyclopentane, methylcyclopentane, cyclohexane, methylcyclohexane, alkenes, OVOCs (excluding MTBE) and aromatics. These mentioned alkanes, aromatics and OVOCs reaction rate coefficients with O₃ are much lower than the alkenes reaction rate coefficients with O₃, which largely counteracted the larger contribution made by the reactions of alkenes and O₃.

13 figures are many. You might want to simplify the manuscript keeping only the most relevant ones in the main body and leave the others to the supplementary information (I suggest to keep 1, 2, 3, 4, (5 could be presented as a table instead of graphically), 6 or 7, 8& 9) Table 1& Table S1. Please readapt these tables to a table/ tables where: concentration, SD, reactivities, reaction coefficients, and refs are included. If the table is too big you can split it in two tables (concentration, SD, reactivities) and reaction coefficients and references. The chemicals should be grouped according to the nomenclature used in the manuscript (BVOC, OVOC…etc.) Supplementary material: Please include some explanations between the figures.

Response: Thanks for the suggestion. We have followed the comments and simplified the manuscript keeping only the most relevant ones in the main body and leave the others to the supplementary information. The tables included concentration, SD, reactivities, reaction coefficients, and refs, and the chemicals were grouped according to the nomenclature used in the manuscript and supplementary material. Some explanations between the figures have been added in supplementary material.

**Reviewer #2**

Oxidation capacity is an important parameter to understand the atmospheric chemistry of air pollutants. This work analyzed the ROH, RO3 and RNO3 based on the measured VOCs and traditional trace gases concentrations in Xiang He. Overall, the methods are reasonable and the data are robust. After the following questions have been well addressed, it is publishable.

Response: we thank the reviewer for the positive comments.

1. Isoprene is also an alkene. I understand the authors want to differentiate the anthropogenic
VOCs from the biogenic VOCs. I suggest to define them more strictly and accurately.

Response: Thanks for the suggestion. Indeed, in this article, we do not distinguish the source of isoprene from biogenic or anthropogenic. The reason is that in urban environment, anthropogenic emission also contribute to isoprene emission (e.g., vehicles ) (Wagner and Kuttler, 2014), so it is hard to distinguish it in the study. Please refer to Line 350-351 in the revised version.

Table S1. The temperature-dependent reaction rate coefficients of trace gases with OH radical, $O_3$ and $NO_3$ radical used in this study.

| Species | Temperature dependence of $k_{OH}$ (cm³ molecule⁻¹ s⁻¹) | Temperature dependence of $k_{O3}$ (cm³ molecule⁻¹ s⁻¹) | Temperature dependence of $k_{NO3}$ (cm³ molecule⁻¹ s⁻¹) |
|---|---|---|---|

| | | | |
|---|---|---|---|
| CH$_4$ | $1.85\times10^{-12}$exp(-1690/T) | $<1\times10^{-23}$ | $<1\times10^{-18}$ |
| Alkanes | | | |
| Ethane | $6.9\times10^{-12}$exp(-1000/T) | $<1\times10^{-23}$ | $<1\times10^{-17}$ |
| Propane | $7.6\times10^{-12}$exp(-585/T)×0.736 | $<1\times10^{-23}$ | $<7\times10^{-17}$ |
| iso-Butane | $1.16\times10^{-17}\times$T$^2\times$exp(225/T)×0.794 | $<1\times10^{-23}$ | $1.06\times10^{-16}$ |
| n-Butane | $9.8\times10^{-12}$exp(-425/T)×0.873 | $<1\times10^{-23}$ | $2.8\times10^{-12}$exp(-3280/T) |
| Cyclopentane | $4.97\times10^{-12}$ | $<1\times10^{-23}$ | $1.4\times10^{-16}$ |
| iso-Pentane | $3.6\times10^{-12}$ | $<1\times10^{-23}$ | $1.62\times10^{-16}$ |
| n-Pentane | $2.44\times10^{-17}\times$T$^2\times$exp(183/T)×0.568 | $<1\times10^{-23}$ | $8.7\times10^{-17}$ |
| 2,2-Dimethylbutane | $3.22\times10^{-11}$exp (-781/T)×0.632 | $<1\times10^{-23}$ | $4.4\times10^{-16}$ |
| 2,3-Dimethylbutane | $1.24\times10^{-17}\times$T$^2\times$exp(494/T)×0.877 | $<1\times10^{-23}$ | $4.4\times10^{-16}$ |
| 2-Methylpentane | $5.4\times10^{-12}$ | $<1\times10^{-23}$ | $1.8\times10^{-16}$ |
| 3-Methylpentane | $5.2\times10^{-12}$ | $<1\times10^{-23}$ | $2.2\times10^{-16}$ |
| n-Hexane | $1.53\times10^{-17}\times$T$^2\times$exp(414/T)×0.061 | $<1\times10^{-23}$ | $1.1\times10^{-16}$ |
| 2,4-Dimethylpentane | $4.77\times10^{-12}$ | $<1\times10^{-23}$ | $1.5\times10^{-16}$ |
| Methylcyclopentane | $5.2\times10^{-12}$ | $<1\times10^{-23}$ | $1.4\times10^{-16}$ |
| 2-Methylhexane | $5.65\times10^{-12}$ | $<1\times10^{-23}$ | $1.5\times10^{-16}$ |
| 2,3-Dimethylpentane | $1.5\times10^{-12}$ | $<1\times10^{-23}$ | $1.5\times10^{-16}$ |
| Cyclohexane | $2.88\times10^{-17}$exp(309/T) | $<1\times10^{-23}$ | $1.4\times10^{-16}$ |
| 3-Methylhexane | $5.6\times10^{-12}$ | $<1\times10^{-23}$ | $1.5\times10^{-16}$ |
| 2,2,4-Trimethylpentane | $3.34\times10^{-12}$ | $<1\times10^{-23}$ | $9.0\times10^{-17}$ |
| n-Heptane | $1.59\times10^{-17}\times$T$^2\times$exp(478/T) | $<1\times10^{-23}$ | $1.5\times10^{-16}$ |
| Methylcyclohexane | $4.97\times10^{-12}$ | $<1\times10^{-23}$ | $1.4\times10^{-16}$ |
| 2,3,4-Trimethylpentane | $6.6\times10^{-12}$ | $<1\times10^{-23}$ | $1.9\times10^{-16}$ |
| 2-Methylheptane | $7\times10^{-12}$ | $<1\times10^{-23}$ | $1.9\times10^{-16}$ |
| 3-Methylheptane | $7\times10^{-12}$ | $<1\times10^{-23}$ | $1.9\times10^{-16}$ |
| n-Octane | $2.76\times10^{-17}\times$T$^2\times$exp(378/T) | $<1\times10^{-23}$ | $1.9\times10^{-16}$ |
| Nonane | $2.51\times10^{-17}\times$T$^2\times$exp(477/T) | $<1\times10^{-23}$ | $2.3\times10^{-16}$ |
| n-Decane | $3.13\times10^{-17}\times$T$^2\times$exp(416/T) | $<1\times10^{-23}$ | $2.8\times10^{-16}$ |
| n-Undecane | $12.3\times10^{-12}$ | $<1\times10^{-23}$ | |
| Alkenes | | | |
| Ethylene | $9.0\times10^{-12}$ (T/300)$^{-0.85}$ | $9.1\times10^{-15}$exp(-2580/T) | $3.3\times10^{-12}$exp(-2880/T) |
| Propylene | $3.0\times10^{-11}$(T/300)$^{-1}$ | $5.5\times10^{-15}$exp(-1880/T) | $4.6\times10^{-13}$exp(-1155/T) |
| trans-2-Butene | $1.01\times10^{-11}$exp (550/T) | $6.64\times10^{-15}$exp(-1095/T) | $3.9\times10^{-13}$ |
| 1-Butene | $6.6\times10^{-12}$ exp(465/T) ×0.87 | $9.64\times10^{-18}$ | $1.35\times10^{-14}$ |
| cis-2-Butene | $1.1\times10^{-11}$ exp(487/T) | $3.22\times10^{-15}$exp(-968/T) | $3.52\times10^{-13}$ |
| 1,3-Butadiene | $1.48\times10^{-11}$exp(448/T)×0.649 | $1.34\times10^{-14}$exp(-2283/T)×0.5 | $1.0\times10^{-13}$ |
| 1-Pentene | $5.86\times10^{-12}$ exp(500/T)×0.87 | $1.06\times10^{-17}$ | $1.5\times10^{-14}$ |
| trans-2-Pentene | $6.7\times10^{-11}$ | $1.6\times10^{-16}$ | $3.7\times10^{-13}$ |
| cis-2-Pentene | $6.5\times10^{-11}$ | $1.3\times10^{-16}$ | $3.7\times10^{-13}$ |
| Isoprene | $2.7\times10^{-11}$exp(390/T) | $1.03\times10^{-14}$exp(-1995/T) | $3.15\times10^{-12}$exp(-450/T) |
| 1-Hexene | $3.7\times10^{-11}$ | $1.31\times10^{-17}$ | $1.8\times10^{-14}$ |
| OVOCs | | | |

| | | | |
|---|---|---|---|
| HCHO | $5.4 \times 10^{-12}\exp(135/T)$ | $<1\times10^{-20}$ | $5.6\times10^{-16}$ |
| Acrolein | 18.3 | $<1\times10^{-20}$ | |
| Propanal | $5.1 \times 10^{-12}\exp(405/T)$ | $<1\times10^{-20}$ | $6.4\times10^{-15}$ |
| Acetone | $8.8\times10^{-12}\exp(-1320/T)+$ $1.7\times10^{-14}\exp(423/T)$ | $<1\times10^{-20}$ | $<3\times10^{-17}$ |
| MTBE | $2.94\times10^{-12}$ | $<1\times10^{-20}$ | |
| Methacrolein | $8.0\times10^{-12}\exp(380/T)$ | $1.4\times10^{-15}\exp(-2100/T)$ | $3.4\times10^{-15}$ |
| n-Butanal | $6.0\times10^{-12}\exp(410/T)$ | $<1\times10^{-20}$ | $1.7\times10^{-12}\exp(-1500/T)$ |
| MethylVinylKetone | $2.6\times10^{-12}\exp(610/T)$ | $<1\times10^{-20}$ | $6.0\times10^{-16}$ |
| Methylethylketone | $1.5\times10^{-12}\exp(-90/T)\times0.462$ | $<1\times10^{-20}$ | |
| 2-Pentanone | $4.4\times10^{-12}$ | $<1\times10^{-20}$ | |
| Pentanal | $6.34\times10^{-12}\exp(448/T)\times0.19$ | $<1\times10^{-20}$ | $1.5\times10^{-14}$ |
| 3-Pentanone | $2\times10^{-12}$ | $<1\times10^{-20}$ | |
| Hexanal | $3.0\times10^{-11}$ | $<1\times10^{-20}$ | $1.6\times10^{-14}$ |
| Aromatics | | | |
| Benzene | $2.3\times10^{-12}\exp(-190/T)\times0.53$ | $<1\times10^{-20}$ | $3.0\times10^{-17}$ |
| Toluene | $1.8\times10^{-12}\exp(340/T)\times0.18$ | $<1\times10^{-20}$ | $7.0\times10^{-17}$ |
| Ethylbenzene | $7\times10^{-12}$ | $<1\times10^{-20}$ | $6.0\times10^{-16}$ |
| m/p-Xylene | $1.89\times10^{-11}$ | $<1\times10^{-20}$ | $2.6\times10^{-16}$ |
| o-Xylene | $1.36\times10^{-11}$ | $<1\times10^{-20}$ | $4.1\times10^{-16}$ |
| Styrene | $5.8\times10^{-11}$ | $1.7\times10^{-17}$ | $1.5\times10^{-12}$ |
| Isopropylbenzene | $6.3\times10^{-12}$ | $<1\times10^{-20}$ | $6.0\times10^{-16}$ |
| n-Propylbenzene | $5.8\times10^{-12}$ | $<1\times10^{-20}$ | $6.0\times10^{-16}$ |
| m-Ethyltoluene | $1.18\times10^{-11}$ | $<1\times10^{-20}$ | $8.6\times10^{-16}$ |
| p-Ethyltoluene | $1.86\times10^{-11}$ | $<1\times10^{-20}$ | $8.6\times10^{-16}$ |
| 1,3,5-Trimethylbenzene | $5.67\times10^{-11}$ | $<1\times10^{-20}$ | $8.8\times10^{-16}$ |
| o-Ethyltoluene | $1.19\times10^{-11}$ | $<1\times10^{-20}$ | $8.6\times10^{-16}$ |
| 1,2,4-Trimethylbenzene | $3.25\times10^{-11}$ | $<1\times10^{-20}$ | $1.8\times10^{-15}$ |
| 1,2,3-Trimethylbenzene | $3.27\times10^{-11}$ | $<1\times10^{-20}$ | $1.9\times10^{-15}$ |
| Criteria pollutants | | | |
| CO | $2.4\times10^{-13}$ | | |
| NO | $3.3\times10^{-11}(T/300)^{-0.3}$ | $1.4\times10^{-12}\exp(-1310/T)$ | $1.8\times10^{-11}\exp(110/T)$ |
| NO$_2$ | $4.1\times10^{-11}$ | $1.4\times10^{-13}\exp(-2470/T)$ | $1.9\times10^{-12}(T/300)^{0.2}$ |
| SO$_2$ | $1.3\times10^{-12}(T/300)^{-0.7}$ | | $<1.0\times10^{-19}$ |
| O$_3$ | $1.7\times10^{-12}\exp(-940/T)$ | | |

Note: The temperature-dependent reaction rate coefficients of VOCs and CO are from Atkinson et al. (1983), Atkinson and Arey (2003), Atkinson et al. (2006), Salgado et al. (2008) and the Master Chemical Mechanism, MCM v3.3.1 via the website: http://mcm.leeds.ac.uk/MCM (last accessed: 25 March 2020); The temperature-dependent reaction rate coefficients of NO, NO$_2$, SO$_2$ and O$_3$ are from Atkinson et al. (2004). T denotes temperature.

Table1. Comparison of speciated OH reactivity with former studies in China.

[revised manuscript text omitted]

5. Traffic is not the only source of NOx. Thus, it is not reasonable to ascribe the ROH to traffic Line 385.

Response: We agree that traffic is not the only source of NOx. However, traffic-related emissions are the main sources of CO and NOx. Thus, We think it is reasonable to ascribe the ROH (OH reactivity) to large influence of traffic-related emissions. However, we have deleted related statements because the comparison of VOCs composition is necessary among different researchers.

6. When comparing the ROH(TVOCs) with other researches, the comparison of VOCs composition is necessary among different researchers (lines 399-419).

Response: The comparison of speciated OH reactivity with former studies in China has been added in the revised version, as shown in Table 1. Please refer to Line 1251-1252 in the revised version.

Table1. Comparison of speciated OH reactivity with former studies in China.

[revised manuscript text omitted]

7. When discussing the implication for control strategies, I think it is more reasonable to normalize the reactivity to secondary pollutants formation potential.

Response: Thanks for the constructive comments. This study was aiming to explore the atmospheric oxidation capacity and photochemical reactivity rather than secondary formation. Therefore, we would like to keep discussing the implication for control strategies based on reactivity. In order to provide recommendations for possible future policies, we have also identified the possible sources of the VOCs whose concentration could be limited based on the certain chemical tracers which are generally presumed to be emitted from specific sources and present in significant amounts in the collected samples. Please refer to Line 567-570 and 584-588 in the revised version.

**Reviewer #3**

This paper shows OH, NO3, and O3 reactivity from VOC and traces gas measurements conducted in Xianghe in 2018 from 6 July to 6 August. In addition, the authors estimate the trace gases oxidation rate using parametrized OH, NO3, and observed O3 concentrations, which is defined as oxidation atmospheric oxidation capacity. This data set helps to add to the increasing knowledge of the oxidant reactivity. The atmospheric oxidation capacity highly depends on the parametrization. Though this method is not new, a detail uncertainty analysis related to the calculation is missing. This reviewer suggests using a box model to calculate the OH and NO3 concentrations or prove the justification of the parameterization. Besides, it's difficult to follow the writing, especially the authors tried to compare their results with other campaigns. The manuscript needs a significant reduction to be concise and informative before reconsidering.

Response: we thank the reviewer for the comments. We think these comments are important to improving the manuscript. According to your comments, a box model SOSAA was used to simulate concentrations of OVOCs, OH and NO$_3$ (Line 244-273 in the revised version). The modeled concentration of OVOCs and observed ones, as well as OH and NO$_3$ concentrations, were compared and discussed (Line 598-616 in the revised version).

Specific comments:

1.Line 266-270, It's not clear which values are used from which literature. If there is difference between different literatures, e.g. OH+NO2, which one is used?

Response: We are sorry for this confusing in the previous manuscript. The related sentences have been revised as follows (Line 285-293 in the revised version):

In the above equations, the temperature-dependent reaction rate coefficients (in cm$^3$ molecule$^{-1}$ s$^{-1}$) for OH-$NMVOC_i$ ($k_{OH+NMVOC_i}$), OH-CO ($k_{OH+CO}$), NO$_3$-$NMVOC_i$ ($k_{NO_3+NMVOC_i}$) and O$_3$-$NMVOC_i$ ($k_{O_3+NMVOC_i}$) are from Atkinson and Are (2003), Atkinson et al. (2006), Atkinson et al. (1983), Salgado et al. (2008) and the Master Chemical Mechanism, MCM v3.3.1 via the website: http://mcm.leeds.ac.uk/MCM (last accessed: 25 March 2020). OH-NO ($k_{OH+NO}$), OH-NO$_2$ ($k_{OH+NO_2}$), OH-SO$_2$ ($k_{OH+SO_2}$), OH-O$_3$ ($k_{OH+O_3}$), NO$_3$-NO ($k_{NO_3+NO}$), NO$_3$-NO$_2$ ($k_{NO_3+NO_2}$), NO$_3$-SO$_2$ ($k_{NO_3+SO_2}$), O$_3$-NO ($k_{O_3+NO}$) and O$_3$-NO$_2$ ($k_{O_3+NO_2}$) are from Atkinson et al. (2004). The temperature-dependent reaction rate coefficients are listed in Table S1 in the Supplementary Materials.

**The correlation between OH and JO$^1$D**

Figure 3 shows the relationship between modeled OH mixing ratio and the measured JO$^1$D. The coefficient of determination (R$^2$) is 0.86, and the linear regression fit shows the slope is $6.1 \times 10^{11}$ cm$^{-3}$ s$^{-1}$ and the intercept is $0.9 \times 10^6$ cm$^{-3}$. These values are comparable to Tan et al. (2017) except the slope is about 36% higher than the observation fit in Tan et al. (2017).

[Figure]

Figure 1: Diurnal mean of modeled (orange solid line) and measured (blue points) mixing ratios of (a) ten calculated and (b) all OVOCs, respectively. The ±1 standard deviation are also shown for modeled (orange shade) and measured (vertical sticks) data.

[Figure]

Figure 2: Modeled diurnal median (solid line) of (a) OH, (b) HO$_2$, (c) RO$_2$ and (d) NO$_3$. The 25th and 75th percentiles are shown as shade.

[Figure]

Figure 3: Correlation between modeled OH number concentration and measured JO[1]D. A linear fit is shown by an orange line, the intercept, slope and $R^2$ values are shown in the legend.

4. The parameterization of NO3 is improved by considering the conversion to N2O5 compared to the first version. A proper discussion related to this uncertainty is missing. In equation (4), AOC is defined as the sum of all trace gases oxidation rate by OH, NO3, and O3. Is NO included? Please declare it clearly.

Response: Thanks for the comment. We have used a column chemical transport model SOSAA to simulate NO$_3$ concentration. Please refer to the responses to the comments #3.

NO is not included in. The term "oxidation capacity" of an oxidant $X$ (= NO$_3$, OH and O$_3$) is defined as the sum of the respective oxidation rates of the molecules $Y_i$ (NMVOCs, CH$_4$ and CO) (Geyer et al., 2001).

$$\text{AOC} = \sum_{i=1} k_{Y_i-X}[Y_i]\,[X] = \sum_{i=1} R_X^{Y_i}\,[X] \quad (4)$$

Here, $[Y_i]$ and $[X]$ are number concentrations of molecule $Y_i$ and oxidant $X$, respectively. $k_{Y_i-X}$ is the temperature-dependent reaction rate coefficients of the molecule $Y_i$ with oxidant $X$. $R_X^{Y_i}$ is the oxidant $X$ reactivity of molecules $Y_i$. Please refer to Line 295-301 the revised version.

Response: We have showed the integral oxidation over a day in Figure 10 (Figure 9 in the revised version).

[Figure]

Figure 9. Comparison of the relative contributions of OH, $NO_3$ and $O_3$ to the 24-h, daytime and nighttime averaged loss rates. Data are calculated for the oxidation of (a, d and g) NMVOCs, $CH_4$ and CO, (b,e and h) NMVOCs only, and (c,f and i) unsaturated NMVOCs only.

6. Figure 11. Why alkenes show a significant variation in RNO3 and RO3 but not ROH?

Response: Thanks for the information. Figure 11 (Figure S12 in the revised version) showed the time series of NMVOCs loss rates due to the reactions with OH radical, $O_3$ and $NO_3$. However, the differences of alkenes variation in OH, $NO_3$ and $O_3$ reactivities can be largely accounted for by the discrepancies of reaction rate coefficients with OH, $O_3$ and $NO_3$. First, the alkenes reaction rate coefficients with $O_3$ and $NO_3$ are much higher than alkanes, aromatics and OVOCs reaction rate coefficients with $O_3$ and $NO_3$. Second, the alkenes reaction rate coefficients with OH are comparable with alkanes, aromatics and OVOCs reaction rate coefficients with $O_3$ and $NO_3$. Third, the orders of magnitude of the differences of alkenes reaction rate coefficients with OH, $O_3$ and $NO_3$ are much smaller than that alkanes, aromatics and OVOCs reaction rate coefficients with OH, $O_3$ and $NO_3$.

7. Figure 12. Maybe it's better to use the same scale for all panels.

Response: We have used the same scale for all panels in Figure 12 (Figure 10 in the revised version).

[Figure]

Figure 10 . Diurnal variations of NMVOCs loss rates due to the reactions with OH radical (blue lines), $O_3$ radical (green lines) and $NO_3$ (red lines).

---

## Author Comment (AC2) · 27 Apr 2020

The comment was uploaded in the form of a supplement:
https://www.atmos-chem-phys-discuss.net/acp-2019-788/acp-2019-788-AC2-supplement.pdf

---

## Author Response (AR2)

Dear editor,

Thank you for handling our submission and we really appreciate your comments to improve the paper!

We also think the reviewer #3 for the fruitful comments and suggestions. In this version, we have revised the manuscript accordingly and addressed all the reviewers' comments point-by-point for consideration as below. The remarks from the reviewers are shown in black, and our responses are shown in blue color. All the page and line numbers mentioned following are refer to the revised manuscript without change tracked.

Yours sincerely,

Yonghong Wang,    yonghong.wang@helisnki.fi
Yuesi Wang        wys@mail.iap.ac.cn

The authors revised the manuscript with improvements, but massive problems, mainly language issue, still exist and need to be revised before publication.

Response: We have revised the manuscript accordingly and addressed the comments and suggestions raised by the reviewer point-by-point. We also corrected some mistakes and contradictions in the context. The Sect. 2.6, Sect. 3.3 and Sect. 3.4.3 have been deleted in the revised version. Besides, the paper has been subjected to editions for language by a native English speaker.

1. As the author replaced the OH and $NO_3$ radical concentrations by model calculation rather than parameterization, the title of the manuscript may not appropriate.

Response: We have followed the comments and the title of the manuscript has been corrected to 'Atmospheric reactivity and oxidation capacity during summer at a suburban site between Beijing and Tianjin'.

2. The reviewer is still missing an argument why all OH, $NO_3$ and $O_3$ reactivity are calculated and compared if OH reactivity dominated 98.2% of the AOC. In the Line 141-144, the authors say previous study has analyzed OH reactivity but not $NO_3$ or $O_3$. Isn't it one of the reasons that AOC is dominated by OH oxidation? In the abstract and conclusion, the authors claimed that the present study provide useful suggestions for VOC pollution control in North China Plain. Maybe the combined integration of OH, $NO_3$ and $O_3$ reactivity analysis could shed light on this topic but it's hidden in the current manuscript.

Response: Hydroxyl (OH), nitrate radicals ($NO_3$) and $O_3$ play a centrally important role in cleansing the atmosphere of trace gas emissions resulting from both anthropogenic and biogenic activity (Atkinson and Arey, 2003;Heard and Pilling, 2003;Lu et al., 2018). OH can react both by addition and H abstraction to organic and inorganic trace gases and may be considered to be more reactive and much less selective than the $NO_3$ radical. The distinct reaction modes lead to significant differences in the lifetimes of both radicals, which for OH are typically less than 1 s and for $NO_3$ can exceed 1 h (Liebmann et al., 2018). The large $NO_3$ mixing ratios at night-time and the large rate constants for the reaction of $NO_3$ with several unsaturated biogenic VOCs result in $NO_3$ being the dominant sink of many BVOCs (Liebmann et al., 2018b;Liebmann et al., 2018a). Although for most NMVOCs, their reaction reaction rate with $O_3$ is much lower than that with either OH or $NO_3$, $O_3$ is very important because it is present at elevated mixing ratios in clean or contaminated atmospheres (Wang et al., 2013). The rate constants of the reactions for some alkenes with $O_3$ are even comparable to those with $NO_3$ (Atkinson and Arey, 2003). As mentioned above OH radicals, $NO_3$ radicals and $O_3$ react with atmospheric trace gases via different mechanisms, resulting in profoundly different rate coefficients and thus reactivities. Therefore, comprehensive evaluations of OH, $NO_3$ and $O_3$ reactivities is a key to understanding atmospheric oxidation capacity and identifying the controlling active species of secondary pollution in the atmosphere. The combined integration of OH, $NO_3$ and $O_3$ reactivities analysis could provide useful suggestions for VOC pollution control in North China Plain. The related statements have been added to the revised context. Please refer to Line 49-51 and 129-133 in the revised version.

**References:**

Atkinson, R., and Arey, J.: Atmospheric Degradation of Volatile Organic Compounds., Chemical Reviews, 103, 4605-4638, doi:10.102/cr0206420, 2003.

Heard, D. E., and Pilling, M. J.: Measurement of OH and $HO_2$ in the troposphere, Chemical Reviews, 103, 5163-5198, doi:10.1021/cr020522s, 2003.

Liebmann, J., Karu, E., Sobanski, N., Schuladen, J., Ehn, M., Schallhart, S., Quéléver, L., Hellen, H., Hakola, H., Hoffmann, T., Williams, J., Fischer, H., Lelieveld, J., and Crowley, J. N.: Direct measurement of $NO_3$ radical reactivity in a boreal forest, Atmos Chem Phys, 18, 3799-3815, doi:10.5194/acp-18-3799-2018, 2018a.

Liebmann, J. M., Muller, J. B. A., Kubistin, D., Claude, A., Holla, R., Plass-Dülmer, C., Lelieveld, J., and Crowley, J. N.: Direct measurements of $NO_3$ reactivity in and above the boundary layer of a mountaintop site: identification of reactive trace gases and comparison with OH reactivity, Atmos Chem Phys, 18, 12045-12059, doi:10.5194/acp-18-12045-2018, 2018b.

Lu, K., Guo, S., Tan, Z., Wang, H., Shang, D., Liu, Y., Li, X., Wu, Z., Hu, M., and Zhang, Y.: Exploring the Atmospheric Free Radical chemistry in China: The Self-Cleansing Capacity and the Formation of Secondary air Pollution, National Science Review, doi:10.1093/nsr/nwy073, 2018.

Wang, Y., Hu, B., Tang, G., Ji, D., Zhang, H., Bai, J., Wang, X., and Wang, Y.: Characteristics of ozone and its precursors in Northern China: A comparative study of three sites, Atmos Res, 132-133, 450-459, doi:10.1016/j.atmosres.2013.04.005, 2013.

3. The manuscript needs proofreading, especially for Introduction section. It is difficult to follow the writing, and there are some obvious mistakes and contradictions in the context. For instance, in line 343-345, why it is 'as expected'? and it said 'with the accumulation of NMVOCs' in the first part of the sentence, but followed by 'the concentration of NMVOCs gradually decreases' in the second part. This sentence is confusing. In line 407, it said 'the majority of OH reactivity towards total NMVOCs values were below 2 s-1', but it was not the case for OH reactivity towards total NMVOCs. The expressions and collocations of some sentences need to be considered again. For instance, in line 610-612, '....., which even dominated the photochemical loss of $NO_3$ '. It is confusing of the 'dominated'.

Response: We have proofread the introduction section and corrected some confusing sentences in the context. We have modified into 'The trend of NMVOCs was inversely related to that of $O_3$. When the NMVOCs concentrations in the atmosphere accumulates to a certain level, as photochemical reactions progress, the $O_3$ concentration gradually increases, and the NMVOCs concentrations gradually decreases (Kansal, 2009;Song et al., 2018)'. Please refer to Line 244-247 in the revised version. We are really sorry for the mistakes in line 407. We have modified into 'The majority of OH reactivity towards total NMVOCs values were below 13 $s^{-1}$ (Figure S4a-d)'. Please refer to Line 310-311 in the revised version. In line 610-612, we have modified into 'The diurnal variation of in the hourly median $NO_3$ concentration showed two peaks which were consistent with the high chemical production from $NO_2 + O_3$ (Figure S9d).'. Please refer to Line 507-508 in the revised version.

**References:**

Kansal, A.: Sources and reactivity of NMHCs and VOCs in the atmosphere: a review, J Hazard Mater, 166, 17-26, doi:10.1016/j.jhazmat.2008.11.048, 2009.

Song, M. D., Tan, Q. W., Feng, M., Qu, Y., Liu, X. G., An, J. L., and Zhang, Y. H.: Source Apportionment and Secondary Transformation of Atmospheric Nonmethane Hydrocarbons in Chengdu, Southwest China, J Geophys Res-Atmos, 123, 9741-9763, doi:10.1029/2018jd028479, 2018.

4. The manuscript still needs a significant reduction to be concise and informative before reconsidering. For instance, the manuscript introduced the $O_3$ formation regime analysis in Sect. 2.6 and Sect. 3.3, but it seems to be irrelevant to the subject and context of this article. The manuscript also analyzed the frequency distributions and cumulative frequency distributions of $k_{OH}$, $k_{O3}$ and $k_{NO3}$, but it not discussed in detail enough nor expressing a clear conclusion.

Response: This work was aiming to explore the OH, $NO_3$ and $O_3$ reactivities based on the measured VOCs and traditional trace gases concentrations. So, following the reviewer's suggestions, the Sect. 2.6 and Sect. 3.3 have been deleted in the revised version. As seen from the frequency distributions and cumulative frequency distributions of total OH, $NO_3$ and $O_3$ reactivities, the majority of total OH, $NO_3$ and $O_3$ reactivities values were dominated by the sum of low reactivity contributions and less by single compounds with high reactivity, highlighting the necessity of

considering a large number of species to obtain a complete picture of total OH, NO$_3$ and O$_3$ reactivities. Please refer to Line 301-304, 369-371 and 417-419 in the revised version.

5. Line 91-93: The OH reactivity is calculated by a global model in Ferracci et al. (2018) and by a box model based on Master Chemical Mechanism in Whalley et al. (2016)

Response: Following the reviewer's suggestions, the sentence of 'The total OH reactivity was also modeled using a zero-dimensional box model based on the Regional Atmospheric Chemical Mechanism to compare them with the measurements or calculations (Lou et al., 2010;Whalley et al., 2016;Ferracci et al., 2018;Yang et al., 2017).' has been modified into 'OH reactivity was also modeled by a global model by (Ferracci et al., 2018) and by a box model based on the Master Chemical Mechanism (MCM) (Whalley et al., 2016).'. Please refer to Line 88-90 in the revised version.

**References:**

Ferracci, V., Heimann, I., Abraham, N. L., Pyle, J. A., and Archibald, A. T.: Global modelling of the total OH reactivity: investigations on the "missing" OH sink and its atmospheric implications, Atmos Chem Phys, 18, 7109-7129, doi:10.5194/acp-18-7109-2018, 2018.

Whalley, L. K., Stone, D., Bandy, B., Dunmore, R., Hamilton, J. F., Hopkins, J., Lee, J. D., Lewis, A. C., and Heard, D. E.: Atmospheric OH reactivity in central London: observations, model predictions and estimates of in situ ozone production, Atmos Chem Phys, 16, 2109-2122, doi:10.5194/acp-16-2109-2016, 2016.

6. Line 325-327, it compared the temperature in this study to the four cities in Tan et al., 2019 without any introduction.

Response: The comparisons of temperature in this study and the four cities (Beijing, Shanghai, Chongqing and Guangzhou) in Tan et al., 2019 were untenable due to the differences of observation seasons. Besides, we have followed the suggestion that the manuscript needs a significant reduction to be concise. Therefore, we decided to delete the related description in the revised version.

7. Figure 10. The y-axis covers 10 orders of magnitude and its difficulty for me to read out the number. It's only mentioned once in the text and not really discussed.

Response: Thanks for the suggestion. First, the similarities and differences between section of overall characteristics of AOC and section of relative contributions of NMVOCs oxidation pathways were confusing because the concept of the oxidation rate is same as AOC. Second, the manuscript needs a significant reduction to be concise. Therefore, we decided to delete the section of relative contributions of NMVOCs oxidation pathways. So, we have moved the Figure 10 and Figures S12-S17 in the revised version.